# Alignment between glioblastoma internal clock and environmental cues ameliorates survival in *Drosophila*

Patricia Jarabo[1], Celia G. Barredo[1], Carmen de Pablo [1,2], Sergio Casas-Tinto [1,2✉] & Francisco A. Martin [1✉]

Virtually every single living organism on Earth shows a circadian (i.e. "approximately a day") internal rhythm that is coordinated with planet rotation (i.e. 24 hours). External cues synchronize the central clock of the organism. Consequences of biological rhythm disruptions have been extensively studied on cancer. Still, mechanisms underlying these alterations, and how they favor tumor development remain largely unknown. Here, we show that glioblastoma-induced neurodegeneration also causes circadian alterations in *Drosophila*. Preventing neurodegeneration in all neurons by genetic means reestablishes normal biological rhythms. Interestingly, in early stages of tumor development, the central pacemaker lengthens its period, whereas in later stages this is severely disrupted. The re-adjustment of the external light:dark period to longer glioblastoma-induced internal rhythms delays glioblastoma progression and ameliorates associated deleterious effects, even after the tumor onset.

[1] Cajal Institute (CSIC), Av Dr Arce 37, 28002 Madrid, Spain. [2]Present address: Drosophila Models for Human Disease Unit, Instituto de Salud Carlos III-IIER, 28220 Madrid, Spain. ✉email: sergio.casas@isciii.es; famartin@cajal.csic.es

Glioblastoma (GB) is a very aggressive brain tumor with no effective treatments, mainly due to glial cells high proliferation, resistance to available treatments, and high mortality rate within the first-year post-diagnosis. Given the obvious clinical interest, a considerable number of animal models mimicking GB has been developed in recent years. In fact, the most widely used model of GB in *Drosophila* recapitulates key aspects of the disease, such as the proliferation of glial cells, invasion, inappropriate differentiation, and related signaling pathways[1]. This model is based on the genetically driven constitutive activation of two signaling pathways (i.e. the activated forms of Epithelial Growth Factor Receptor -EGFR- and PI3K) in all glial cells, thus transforming the whole glial population[2]. The Collection of Somatic Mutations in Cancer database (COSMIC) indicates that mutations in *PI3K* and *egfr* genes are among the 20 most frequent mutated genes found in GB patients (i.e. Astrocytoma grade IV), showing a corresponding frequency of 9% and 14%. Other components of PI3K and EGFR pathways are also commonly mutated, such as *pten* (22%) and *NF1* (11%), respectively.

GB patients show a wide range of neurological symptoms which depends on the localization of the tumor and its expansion[3–5]. It has been historically assumed that these symptoms were the consequence of intracranial pressure and the subsequent edema[6]. However, recent work in different animal models suggests that GB-induced progressive neurodegeneration may be the main reason behind such neurological symptoms[7,8]. For instance, there is a significant decrease in synapse number in the motor neurons of the *Drosophila* model upon GB induction[8].

Although molecular and cellular mechanisms that control synapse number are not completely known, several signaling pathways modulate the number of functional synapses (also known as active zones) in *Drosophila*[9,10]. Among them, the most notorious is Insulin signaling: the activation of the pathway increases the number of active zones, whereas its inactivation decreases it[9,10]. These functions are conserved in the mammalian brain, at least for some of the components[11,12]. Our previous work showed that ImpL2, an antagonist of the Insulin pathway, is required for GB development and its over-expression in glial cells induces a synapse reduction similar to what it is seen in GB brains[13]. Moreover, we also demonstrated that the expression of pro-synaptogenic factors had neuroprotective effects attenuating neurological deficits and lethality not only in GB but also in a *Drosophila* neurodegenerative disease model mimicking Alzheimer's disease[14,15].

The synapse number of particular circuits also changes following the regular circadian rhythmicity in most organisms, including *Drosophila* and vertebrates[16]. Circadian (i.e. "around a day") rhythms refer to the adaptation of life to the daily rotation of the Earth, something that causes repetitive environmental changes every 24 h. Virtually all living organisms coordinate their biological processes with these periodic fluctuations thanks to an endogenous circadian pacemaker located in the brain. This central clock modulates systemically many different physiological processes (reviewed in[17]). The pacemaker is synchronized with the external timing throughout several environmental cues such as light, feeding and social interactions, among others[17]. But, even in the absence of any external cue, physiological processes under circadian control oscillate with a 24 h period. This is achieved by two means. It uses a molecular clock machinery composed by two evolutionary conserved protein heterodimers (PER/TIM and CLOCK/CYCLE in *Drosophila*, with their corresponding mammalian counterparts PER/CRY and CLK/BMAL1). The mechanism involves a negative transcriptional feedback loop with a near-to-24 h oscillation period[17]. Besides, this molecular clock coordinates the activity of a cellular network that controls

systemically the circadian-related functions. In *Drosophila* it is composed by approximately 150 neurons divided into seven main clusters, named according to their anatomical location[18]. Specific subsets of neurons are responsible for different aspects of circadian behavior. Among them, large and small ventral lateral neurons (l-LNv and s-LNv, respectively) express the neuropeptide *pigment-dispersing factor* (*pdf*). PDF is the critical neuropeptide of LNv and the main regulator of activity/rest cycles in the fly[18].

Disruption of circadian rhythms increases the risk to develop different disorders and diseases, including cancer (reviewed in ref. [19]). However, the mechanisms that tumors use to modify biological rhythms, and more specifically the circadian pacemaker, are much less known. Here, we study the interaction between GB and the timekeeping system. Data show that the tumor-induced neurodegeneration also impairs circadian rhythms, mainly because it diminishes the number of synapses. Indeed, GB progressively lengthens the period and ultimately causes arrhythmicity. The adjustment of the external Light:Dark period to a longer one (thus closer to the endogenous period of GB-induced flies) reduces GB growth, rescues GB-induced lethality, and leads to a longer lifespan. Also, if animals already bearing an advanced GB are subjected to a 14:14 h Light:Dark regime (instead of 12:12 h), both GB progression and GB-induced neurodegeneration are significantly slowed down. Would these observations and their mechanisms be conserved in humans, new therapeutic approaches might be proposed in order to improve the life quality of GB patients.

## Results

**GB flies show disrupted circadian rhythms due to neurodegeneration.** To study whether or not *Drosophila* GB model exhibits alterations in biological rhythms, we used *Drosophila* Activity Monitors (DAMs) for the analysis of circadian-associated locomotor activity patterns. The strength of rhythmicity is measured by the power (the height of the periodogram peak) and it gives the significance of the calculated period (see M&M). We established a threshold to distinguish between rhythmic flies (power above 40) and arrhythmic (below 40).

After one-day recording under Light:Dark (L:D) conditions, flies were subjected to a Dark:Dark (D:D) regime, given that the circadian pacemaker is able to maintain biological rhythms in the absence of any external cues[17]. Animals bearing a GB within the first five days of tumor development (early glioma) show few signs of neurodegeneration, brain overgrowth, or movement alterations[13]. Control flies were strongly rhythmic, as shown by an actogram depicting the locomotor activity of one single fly that was representative (see Fig. 1a). Results showed that the average total power in control animals was 248 ± 9.8 and the mean period was 23.7 ± 0.05 h, as expected (Table 1). In contrast, 38% of early GB animals were arrhythmic, with power values under 40. Figure 1b, c illustrates two representative actograms of single early GB flies as examples of rhythmic or arrhythmic animals. Despite high data dispersion and weak rhythmicity made a proper circadian analysis difficult to do, GB rhythmic flies had the tendency to lengthen the period, showing a mean of 25.2 ± 1.2 h, whereas the average power was 80 ± 14.5 (Table 1). In the case of animals with 5 to 10 days of GB development (medium glioma), 87% of the flies showed arrhythmic behaviors. Actually, their average power was 16.1 ± 8.3 whereas control animals show a mean of 150.1 ± 15.2 (Table 1). The average period of a few rhythmic animals was 28.6 ± 4.2 h. Most animals with 10 to 15 days of tumor growth (late GB) died during the analysis, making it impossible to quantify circadian parameters. In Fig. 1k all power values are depicted together with the median, thus showing the great variability of early GB and the low power of medium GB.

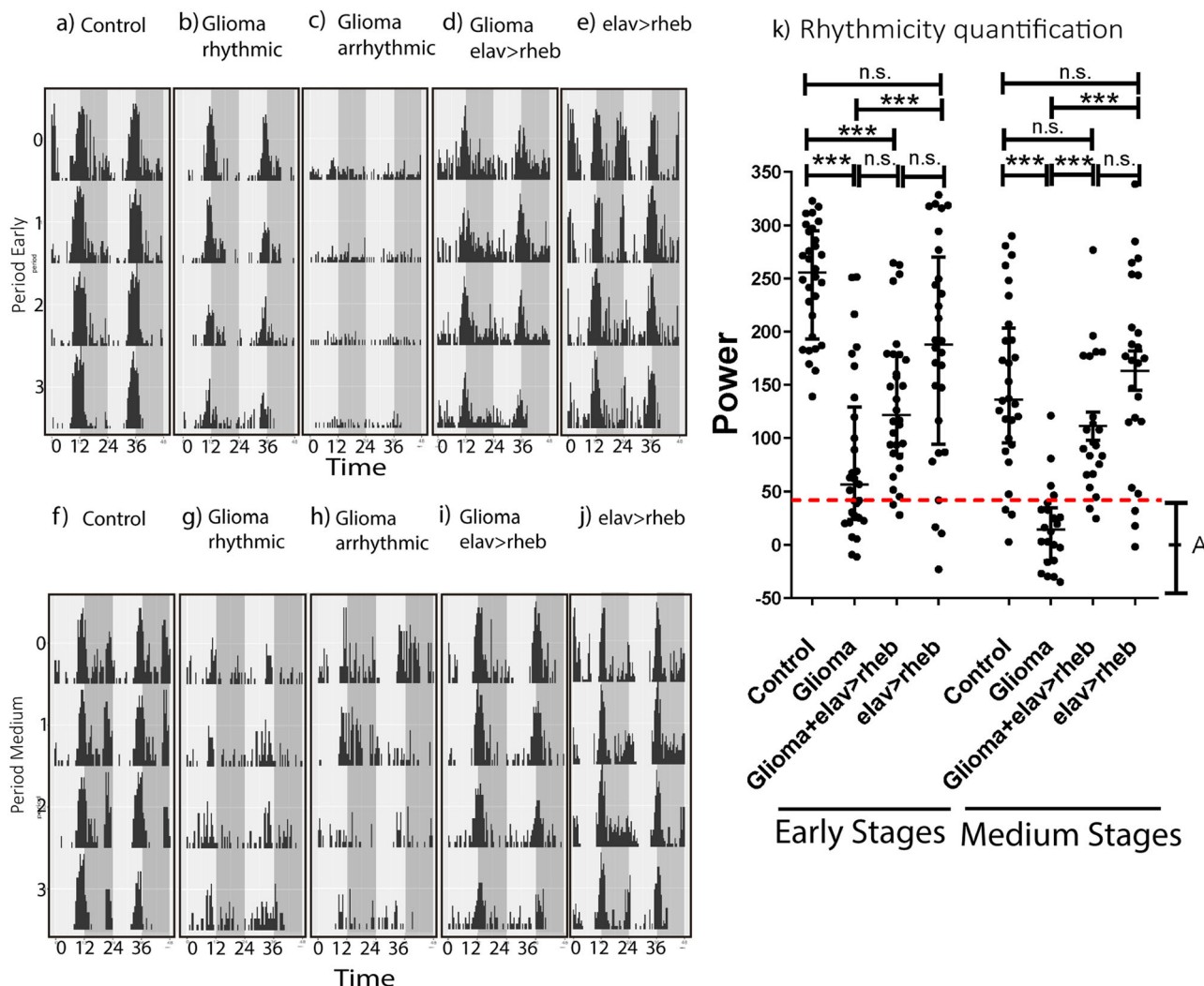

**Fig. 1 The rescue of glioma-induced neurodegeneration restored circadian rhythms.** Representative individual locomotor activity profile from one particular animal of each indicated genotypes showing 5 days under free running (D:D) conditions for (**a**, **f**) control (**b**, **c**, **g**, **h**) glioma, (**d**, **i**) rescued glioma (by means of neuronal *rheb* over-expression) and (**e**, **j**) rescued control flies. Gray bars indicate subjective night. For GB two different actograms are depicted, exemplifying rhythmic and arrhythmic animals. For early GB (**a**–**e**), the locomotor activity of flies during the first 5 days of tumor development was recorded using DAMs, whereas for medium GB (**f**–**j**), the movement of animals from day 5 to day 10 of GB growth was studied. **k** A graph representing the individual power for each fly under different conditions, depicting the median, from at least n > 22 surviving flies (ANOVA, post-hoc Bonferroni) (**P-value < 0.01, ***P-value < 0.001). Values below the dotted line were considered arrhythmic (i.e. A).

**Table 1 Table showing the main circadian values from controls, glioma and rescued glioma flies.**

| | | Total power | | | Rhythmicity | Period | |
|---|---|---|---|---|---|---|---|
| Early (0d–5d) | *n* | Mean | Median | SEM | | Mean | SEM |
| Control | 30 | 248.0 | 255.3 | 9.8 | 100% | 23.7 | 0.1 |
| Glioma | 29 | 80.2 | 56.5 | 14.5 | 62% | 25.2 | 1.3 |
| Glioma+elav>rheb | 30 | 133.4 | 121.4 | 12.4 | 93% | 24.1 | 0.1 |
| elav>rheb | 28 | 183.4 | 188.1 | 19.8 | 86% | 23.7 | 0.1 |
| Medium (5d–10d) | | | | | | | |
| Control | 28 | 150.1 | 136.1 | 15.2 | 86% | 23.7 | 0.1 |
| Glioma | 22 | 16.1 | 14.2 | 8.3 | 14% | 28.6 | 4.3 |
| Glioma+elav>rheb | 22 | 111.2 | 94.5 | 13.7 | 81% | 24.0 | 0.2 |
| elav>rheb | 24 | 163.3 | 174 | 18.7 | 88% | 23.8 | 0.1 |
| Medium (5d–10d) | | | | | | | |
| Control | 27 | 202.6 | 231.6 | 16.9 | 93% | 23.8 | 0.0 |
| Glioma | 16 | 111.3 | 114.4 | 20.5 | 81% | 24.9 | 1.1 |
| Glioma+pdf>rheb | 21 | 93.5 | 74.5 | 19.1 | 76% | 24.7 | 0.7 |

GB-induced neurodegeneration is caused by reduced neuronal insulin signaling and viability was partially rescued by increasing insulin signaling in neurons[13]. We wondered if this approach could be used to experimentally rescue GB effects on circadian parameters. In order to do so, we employed the *LexA/LexAOp* binary expression system combined with the classical Gal4/UAS (see the M&M). The over-expression of *LexAOp-rheb* in virtually all neurons (by means of an *ELAV-LexA* driver) increases insulin activity and consequently blocks GB-induced neurodegeneration[13]. GB and *ELAV > rheb* rescue were induced simultaneously. These animals were all rhythmic with a 24 h mean endogenous period (see Fig. 1d for a representative actogram) and the rhythmicity strength was substantially increased when compared to early GB developed animals, reaching an average power value of 133.4 ± 12.4 (Table 1). However, this ELAV-rescued flies still show reduced rhythmicity compared to control flies, as expected considering the partial rescue of their lifespan[13]. Results were similar in a relatively advanced phase of tumor growth (5–10 days—see Fig. 1i), with mean power values closer to control ones than to medium GB flies (111.2 ± 13.7) and a normal period of 24 ± 0.2 (Table 1). This is also evident in Fig. 1k: *ELAV > rheb*-rescued animals show significant power differences with GB but not with control flies. In contrast, both early and medium control *ELAV > rheb* flies behaved rhythmically, with a 23.7 h and 23.8 h endogenous period, respectively (see Fig. 1e, j for illustrative actograms of each phase). In addition, the average power value was high, with a mean of 183.4 ± 19.8 for early flies and 163.3 ± 18.7 for medium ones (Fig. 1k and Table 1).

In summary, the activation of insulin signaling in neurons prevents neurodegeneration and rescues circadian disruptions caused by GB progression.

**GB reduces synapse number in PDF-expressing neurons**. The role of insulin signaling in the synaptogenic pathway was described almost exclusively in the classical neuron-muscle synapse NMJ (neuromuscular junction) model[8,14,20]. However, the rescue of GB-caused alterations on biological rhythms by means of *rheb* over-expression strongly suggests that it also affects adult neuron-neuron synapses. We focused on s-LNv, the circadian pacemaker neuronal cluster that expresses *pdf* (the main circadian neuropeptide)[17]. We quantified synaptic contacts of sLNv dorsal terminals using a specific antibody against Bruchpilot (Brp), a key component of the presynaptic specialization (i.e. active zones). Brp was previously used to study the circadian plasticity of PDF-neuronal axons[21,22]. We overexpressed either pro- or anti-synaptogenic factors from the insulin pathway (PI3K or GSK3, respectively) in PDF neurons. The quantification of active zones of brains at 7 days post-GB induction showed a statistically significant difference in both cases, although in opposite directions (Fig. 2a–d): activating insulin pathway (PI3K) increased the number of active zones whereas lowering insulin activity (GSK3) reduced its number.

We quantified the number of active zones in PDF neurons of GB brains at day 7 after tumor induction. The results show a significant reduction in the number of synapses (Fig. 2e–g). Modifications of synaptic contacts should also reduce PDF-positive puncta, given the inter-dependence between the number of active zones and PDF protein levels: modifying either one affects the other[23]. To determine whether *pdf* expression was altered upon GB induction, we performed qPCR of medium-developed GB brains and analyzed *pdf* expression levels. qPCR results showed a reduction of *pdf* expression to one third in GB animals, thus suggesting that the reduction of *pdf* expression might be the mechanistic cause of circadian alterations (Fig. 2h).

Beyond *pdf*, we analyzed the expression of other circadian genes. Our results showed almost no differences of *tim/per* and

*clock/cycle* gene expression levels upon GB induction, with the exception of *clock* expression that shows upregulation (Fig. 2h). However, the functionality of each heterodimer demands a perfect stoichiometry for both components[18]. Although we could not rule out a post-translational effect, in terms of expression, the components of the central negative feedback loop that sets the pace of the clock seemed to be operative in the GB brains; however, the main synchronizer of the neuronal network, PDF, was severely affected.

To confirm this hypothesis, we compared adult GB and normal brains stained with an anti-PDF antibody. We noticed that the intensity of the inmuno-fluorescence was weaker in GB-induced brains (Fig. 2j). Actually, there was a significant reduction in the number of PDF puncta in GB brains, in line with the decrease of synapse number (Fig. 2i, j, m). We reasoned that *rheb* over-expression in PDF neurons *(pdf-LexA/LexAOp-Rheb)* should restore PDF levels/active zones. Again, we made use of the *LexA/ LexAOp* binary expression system *(pdf-LexA/LexAOp-Rheb)*. PDF-rescued GB brains tended to increase the number of PDF puncta when compared to GB samples, showing no significant differences with control brains (Fig. 2k–m). Control flies that overexpressed *rheb* in PDF neurons had higher number of active zones/PDF contacts than control and PDF-rescued GB samples; however, the difference was not significant. These results suggest that insulin pathway activation can partially rescue the loss of active zones in GB-induced brains. To recapitulate, both *pdf* expression and function were affected by the GB. In addition, PDF neurons responded to GB-caused neurodegeneration and to the activity of Rheb acting as a synaptogenic signal. Altogether, our data indicate that Insulin signaling contributes, but is not sufficient, to rescue *pdf* expression/rhythmicity.

**GB-induced neurodegeneration in PDF neurons causes circadian alterations**. We set to determine the consequences of preventing neurodegeneration of the central clock (i.e. PDF neurons) upon the biology of GB animals. Due to the progressive nature of the neurodegenerative process and the intrinsic unpredictability of GB development, we found temporal variability in these assays[8,13]. In particular, despite medium-staged (i.e., 5 to 10 days of tumor development) control and GB animals were both rhythmic, their power was significantly reduced in GB flies when compared to control (111.3 ± 20.5 vs 202.6 ± 16.9, Fig. 3a–d and Table 1). The period is lengthened about one hour, up to 24.9 ± 1 h (Table 1). Unexpectedly, PDF-rescued GB animals showed similar values regarding power and period as in flies bearing a glioma (93.5 ± 19 and 24.7 ± 0.7 h, respectively), thus suggesting circadian rhythms were not effectively rescued. Indeed, statistical analysis showed that PDF-rescued flies did not have significant power differences compared to GB flies but they did in comparison with control flies (Fig. 3d). In fact, the shorter lifespan caused by the GB was not rescued by impeding neurodegeneration specifically in PDF neurons (Fig. 3e). To summarize, restoring insulin signaling only in *pdf*-expressing neurons (and consequently, increasing synapse number) was insufficient to reestablish complete normal circadian rhythms or survival rates.

Despite sLNVs determine the period, they still need a functional network of different circadian clusters to establish rhythmicity. We also noticed that during GB progression the locomotor activity is compromised, more likely by the neurodegeneration of motor and sensory neurons. In order to identify when locomotor deficits started, we determined locomotion performance of GB-induced and control animals by using a classical negative geotaxis assay (see M&M). In brief, we quantified the number of flies that reached a 4-cm threshold line after shaking. Animals bearing GB started to show a reduced

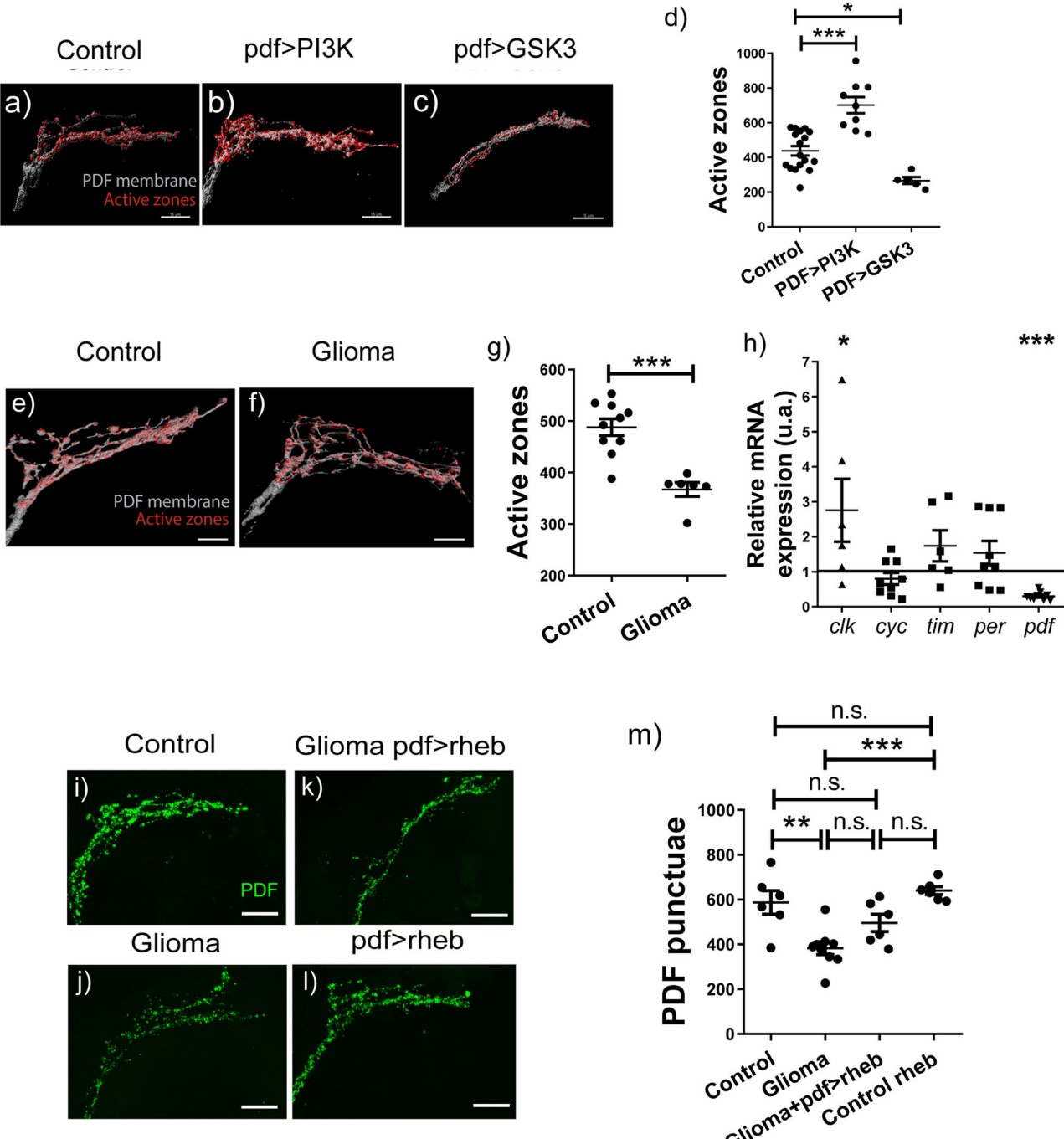

**Fig. 2 GB-induced neurodegeneration of central pacemaker neurons. a–c** Digital reconstruction of PDF neuron axonal terminal membrane of PDF neurons contacting DN1 neurons of (**a**) control, (**b**) PDF > PI3K and (**c**) PDF > GSK3. Axon terminal in white and active zones localized in their membrane in red (scale bar, 15 μm). **d** Quantification and statistical analysis of active zones per animal in at least $N = 5$ flies (ANOVA, post-hoc Bonferroni) (***$p$-value< 0,001). **e**, **f** Digital reconstruction of the axonal terminal membrane of PDF neurons contacting DN1 neurons of adult brains of (**e**) control and (**f**) glioma animals 7 days after tumor induction, with neuronal membrane in white and active zones in red. Digital reconstruction of PDF neurons axon terminal in white and active zones localize in their membrane in red (scale bar, 10 μm). **g** Quantification of active zones per animal for at least $N = 6$ flies (t test) (***$p$-value< 0.001). **h** RT-qPCR analysis of complete brains of adult flies (bearing a 7-day old GB) from control and glioma genotypes in L:D conditions at ZT6 for the circadian genes *clk*, *cyc*, *tim*, *per* and *PDF*, with at least 6 replicates per genotype (t test) (*$p$-value<0,05, **$p$-value<0,01, ***$p$-value<0,001). **i–k** Confocal images of PDF-neuronal contacts with DN1 stained with anti-*pdf* (green) of (**i**) control, (**j**) glioma, (**k**) rescued glioma (by means of *rheb* over-expression in PDF neurons) and (**l**) rescued healthy flies. **m** Quantification and statistical analysis of active zones per animal in at least $N = 6$ flies (ANOVA, post-hoc Bonferroni) (**$p$-value< 0,01).

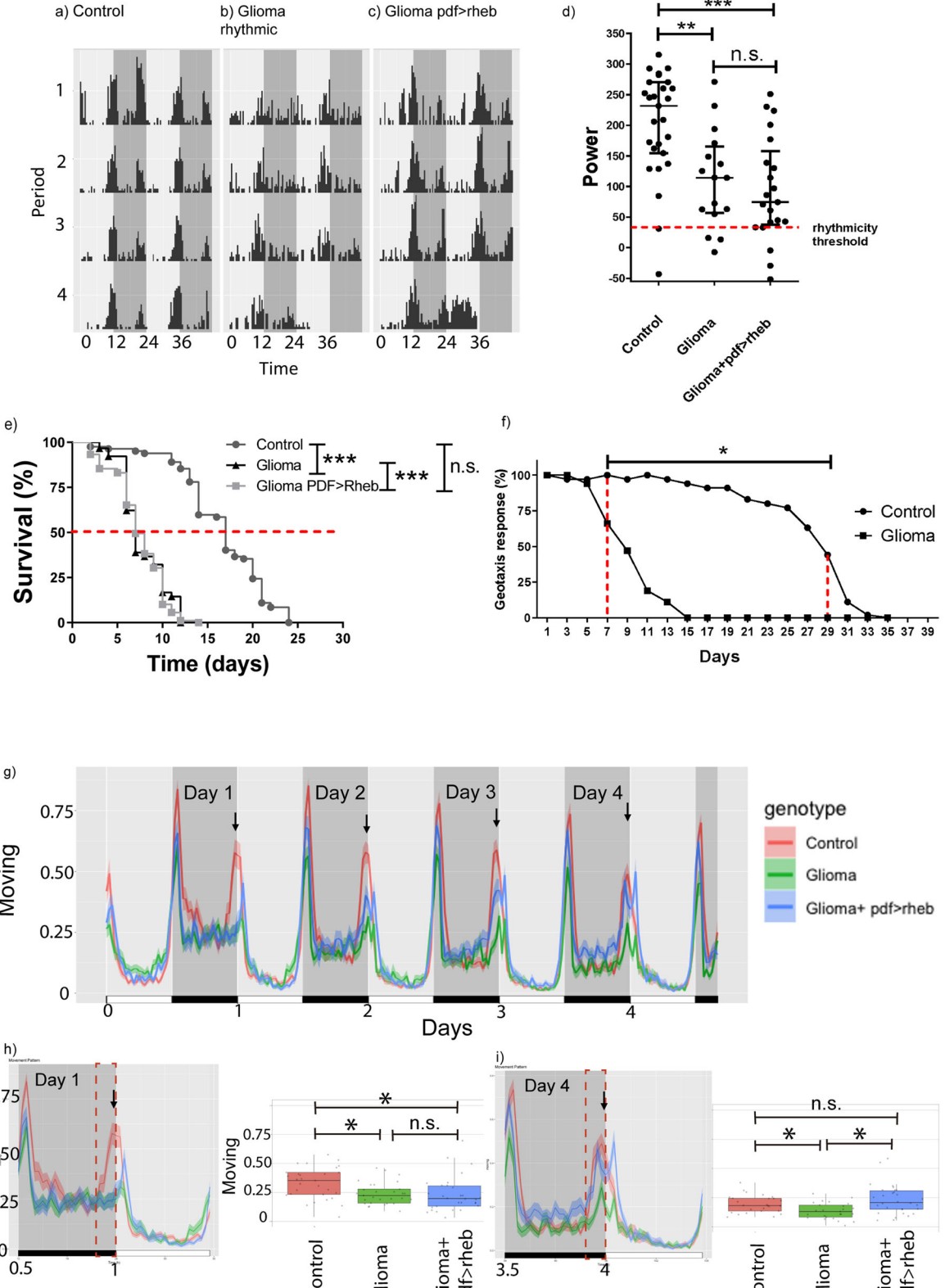

movement at day 7 after tumor induction (Fig. 3f). As a consequence, we focused on early stages of GB development to avoid collateral effects from diminished motility.

The morning anticipation (MA, in our case defined as the increased locomotion during 3.5 h before dawn) depends almost exclusively on the input of PDF-positive s-LNv[24]. L:D conditions allow to study the central pacemaker properties by focusing in the MA. In GB flies MA is lost when compared to control animals (arrows in Fig. 3g). Moreover, the rescue of neurodegeneration by *rheb* over-expression in s-LNV restored this anticipation under

**Fig. 3 Rescue of GB-induced neurodegeneration in PDF neurons does affect GB-induced morning anticipation.** Representative individual locomotor activity profile from one particular animal of each indicated genotypes showing 5 days under free running in D:D conditions for (**a**) control, (**b**) medium glioma and (**c**) rescued glioma (by means of PDF-neuronal *rheb* over-expression) flies. The movement of animals from day 5 to day 10 of GB growth was recorded using DAMs. **d** A graph representing the individual power for each fly under the different conditions, at least $n > 16$ surviving flies (ANOVA, post-hoc Bonferroni) (**p-value < 0.01, ***p-value < 0.001). **e** A survival assay of the different genotypes with statistical analysis. $N > 80$ flies (Mantel–Cox test) (*p-value < 0.05, ** p-value < 0.01, ***p-value < 0.001). **f** Time course of the negative geotaxis assay of control and glioma flies. Data are shown as the percentage of animals that reached a 4-cm line in a tube. $n = 30$ flies (Mann–Whitney $U$ test)((*p-value< 0.05 within the shown interval). **g** The locomotor activity of control, glioma and PDF-rescued glioma flies for the first 5 consecutive days of tumor development (at least $n = 16$ animals). Arrows indicate the morning anticipation (increased activity in the 3.5 h previous to lights on), which is lost in glioma flies and progressively rescued to almost control levels by *rheb* over-expression in PDF neurons. **h, i** Detail of day 0.5-day 1.5 and day 3.5 and 4.5, showing no rescue and an almost full rescue of morning anticipation, respectively, together with the statistical analysis of the morning anticipation in day 1 and day 4. At day one, there is a significant difference between control and rescued glioma animals, comparable to glioma flies. At day 4; however, rescued glioma animals are more similar to control ones and significative different from glioma flies (Pairwise comparisons using Wilcoxon rank sum test) (*p-value < 0.05).

L:D conditions (Fig. 3g). Interestingly, our results show that the rescue effect improved progressively: in the first morning after GB (and PDF-rescued) onset, the MA of GB and PDF-rescued GB flies was indistinguishable and showed a significant difference with control animals; in contrast, the fourth day showed a peak of anticipation similar to control animals that is significantly different to glioma flies (Fig. 3h, i). In summary, impeding neurodegeneration in PDF neurons rescued their functionality related to the morning anticipation.

**Re-adjusting the external environment rescues GB-induced effects.** Flies bearing a GB tended to lengthen their period before reaching the typical arrhythmicity caused by the tumor (Table 1). Given the impossibility to adapt the external period to each individual animal and the fast (and lethal) development of the glioma in the fly, we decided to analyze the GB behavior under a subjective day of 14 h of light and 14 h of darkness for all the flies. Results showed that 7 days after GB induction the number of cells in the brain of animals under a 14:14 L:D regime was comparable to a control brain without a tumor (Fig. 4a–f), although not in total volume (quantified by measuring membrane, see M&M). In parallel, animals raised under a 24 h period and bearing a GB showed a significant difference both in number of cells and volume of glial cells compared with control brains (Fig. 4a-f). This was also evident when we quantified the number of active zones in adult NMJs: animals with a GB after 7 days under 14:14 L:D conditions showed a similar number of synapses as in their control counterparts, whereas the number of active zones was clearly diminished in the case of GB brains in a regular 12:12 L:D period (Fig. 4g–k).

Next, we addressed the impact on survival that 14:14 L:D routine would have in GB animals. GB flies showed a reduced lifespan compared to controls under a regular L:D regime. In contrast, GB animals under 14:14 L:D conditions had longer lifespan related to its corresponding control (Fig. 4l). Intriguingly, we noticed that the 14:14 L:D cycle also increased the lifespan of wild type flies, so to illustrate the rescue under the new regime we represented survival (i.e. p50) as a percentage of the time when half of GB flies were still alive respect to the time when the same proportion of control animals was dead (dotted red line in the graph). Under normal conditions of 12:12 L:D, the survival of GB-bearing animals was 57% of the control, whereas under a 14:14 L:D regime the survival increased up to 78%, being the difference quite statistically significant (Fig. 4m). Basically, the re-adjustment of the external cues to a longer L:D period was sufficient to rescue the phenotypes associated to a GB development, including survival.

**Re-adjusting environmental cues after GB induction also reduces neurodegeneration.** Finally, we wondered if shifting from a normal routine 12:12 L:D to a 14:14 L:D regimen would have an impact on GB progression and/or neuronal function after the GB condition is underway. In our hands, inducing a tumor for 7 days is sufficient to detect an effect on neurodegeneration and glial cell number[13]. Therefore, 4 days after GB induction under normal conditions we transferred GB flies to a 14:14 L:D regimen. Results showed a significant decrease in the number of glial cells after 4 days of tumor development under 14:14 L:D treatment when compared with GB animals maintained at 12:12 L:D (Fig. 5a–c). In line with results included in Fig. 4, the membrane of the GB was enlarged in treated tumoral brains (i.e. shifted to 14:14 L:D conditions) at similar levels as in GB under 12:12 L:D (Fig. 5d). Moreover, when we searched for neurodegeneration evidences (like synapse loss in NMJs), we did observe that treated GB brains had substantially more active zones that GB brains under normal L:D conditions (Fig. 5e–g). In conclusion, GB-associated phenotypes could be alleviated by modifying the environmental cues, even after the tumor onset.

## Discussion

The current *Drosophila* model of GBM does not reproduce the initial steps of GBM formation, given that all glial cells are transformed (leaving no healthy glia), so the interaction between healthy and tumoral glial cells cannot be assessed. GBM cells are able to activate astrocytes in order to induce invasion in the healthy tissue[25]. Therefore, healthy glial cells promote GBM migration, invasion, proliferation and angiogenesis[26]. Unfortunately, whereas in a mammalian brain 50% of cells are glia, in *Drosophila* the percentage is 10%[27], thus suggesting that this *Drosophila* GBM model might not be adequate to study GB-healthy glia interaction. In addition, GBMs typically arises from a single cell in a clonal manner, with a strong spatio-temporal developmental pattern[28,29], feature that is not reproduced in this *Drosophila* model.

However, it reproduces faithfully the diffuse tumor phase found in well-developed human glioblastomas. Actually, our intention using this *Drosophila* GB model is to study neurological defects caused by the diffuse front of GBM even after treatments such as surgery. In this scenario, GBM cells are distributed throughout the brain without generating a big core mass, indeed similar to what we see in *Drosophila* GBM. For instance, this model was validated regarding many of advanced human disease features, such as neurodegeneration, signaling pathways involved (like vesicular trafficking, YAP/TAZ and metabolic genes) and eventual lethality[1,30–32]. Recently, we showed that ImpL2-induced low Insulin signaling in neurons was responsible for the neuro-degenerative phenotype in a *Drosophila* adult GB model, and that increasing neuronal insulin function alleviated GB-induced effects[13]. Building on this previous work, we now extend our study to the mechanisms that GB uses to modify circadian rhythms. We found that neuronal insulin activity rescues GB-

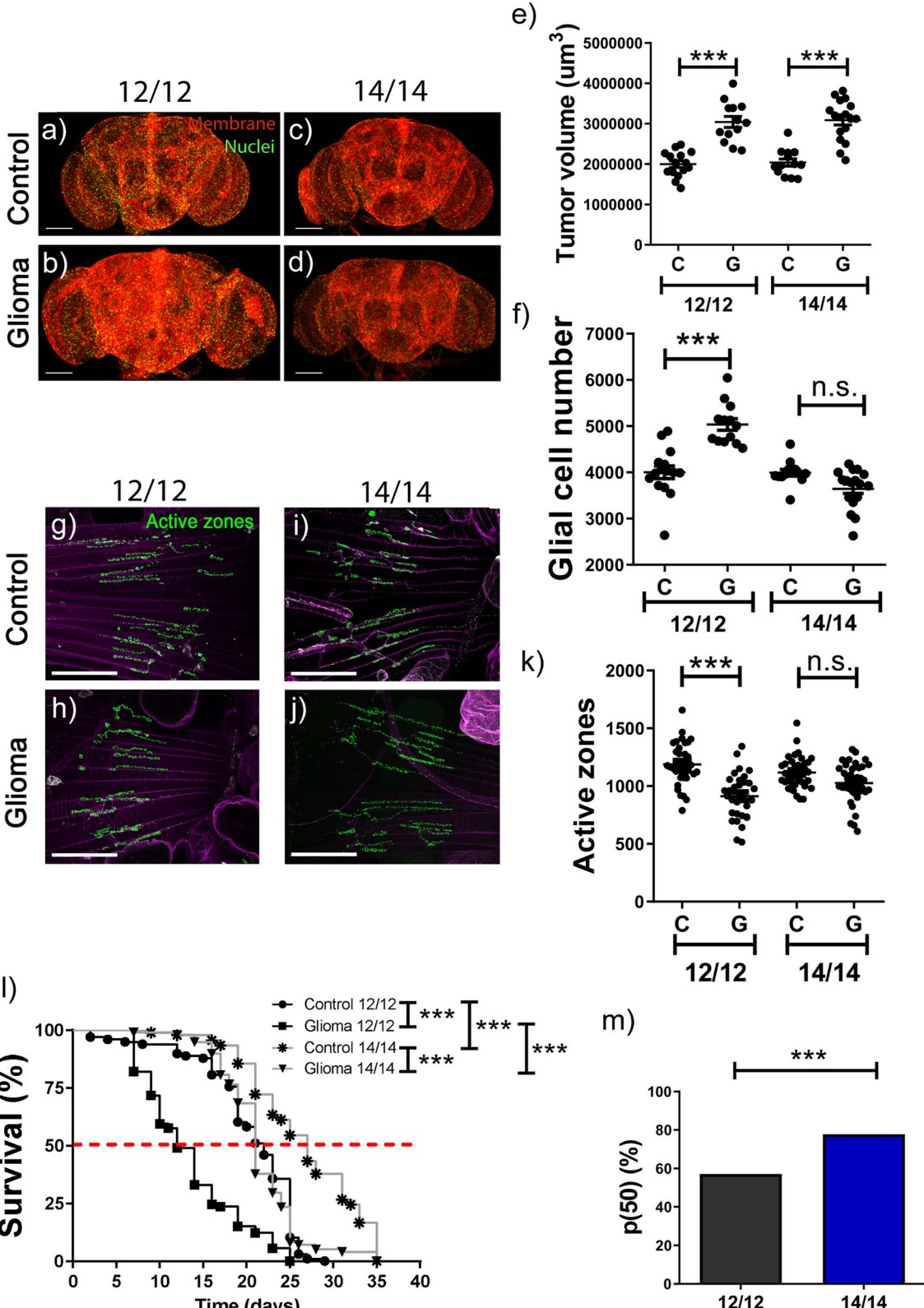

induced circadian disruptions and, in fact, PDF neurons showed altered *pdf* expression and function. Restoring insulin signaling (thus increasing synapse number) in such PDFn rescued morning anticipation, a specific circadian response, despite it was not able to rescue normal circadian rhythms or survival. Re-alignment of external cues to a longer L:D period effectively rescued GB-associated phenotypes, even after the GB onset.

The epidemiological relation between disrupted circadian rhythms and cancer is well documented[33]. Recent work shed light on possible molecular mechanisms connecting the disturbed autonomous circadian clock of the tumor with the regulation of metabolism, cell cycle, DNA repair machinery and apoptosis -mostly mediated by Myc-[34]. Remarkably, disruptions in the central clock also play an important role in tumor appearance and

**Fig. 4 Synchronization of environmental light/dark phases with internal period prevents GB-induced neurodegeneration. a–d** Confocal microscopy images of the entire adult brain from control and glioma flies under (**a**, **b**) 12:12 and (**c**, **d**) 14:14 L:D conditions. Glial membrane is tagged in red and glial nucleus in green. **e**, **f** Quantification of glial cell number and glial membrane volume of the whole brain per animal for at least $N > 12$ flies (ANOVA, post-hoc Bonferroni) (scale bar, 100 μm) (***$p$-value < 0,001). **g–j** Confocal microscopy images of adult NMJ from control and glioma flies under 12:12 (**g**, **h**) and 14:14 (**i**, **j**) L:D conditions. Active zones are tagged in green. **k** Quantification of active zones per animal, $N > 30$ flies (ANOVA, post-hoc Bonferroni) (scale bar, 50 μm) (***$p$-value<0,001). **l** Graph shows the lifespan of control and glioma animals under 12:12 and 14:14 L:D conditions (Mantel–Cox test for $N > 90$ flies) (***$p$-value < 0,001). **m** p50 of glioma flies at 12:12 or 14:14 L:D, showing a significant difference. p50 is defined as the time when half of GB flies are still alive respect to the time when the same proportion of control animals are dead, in percentage.

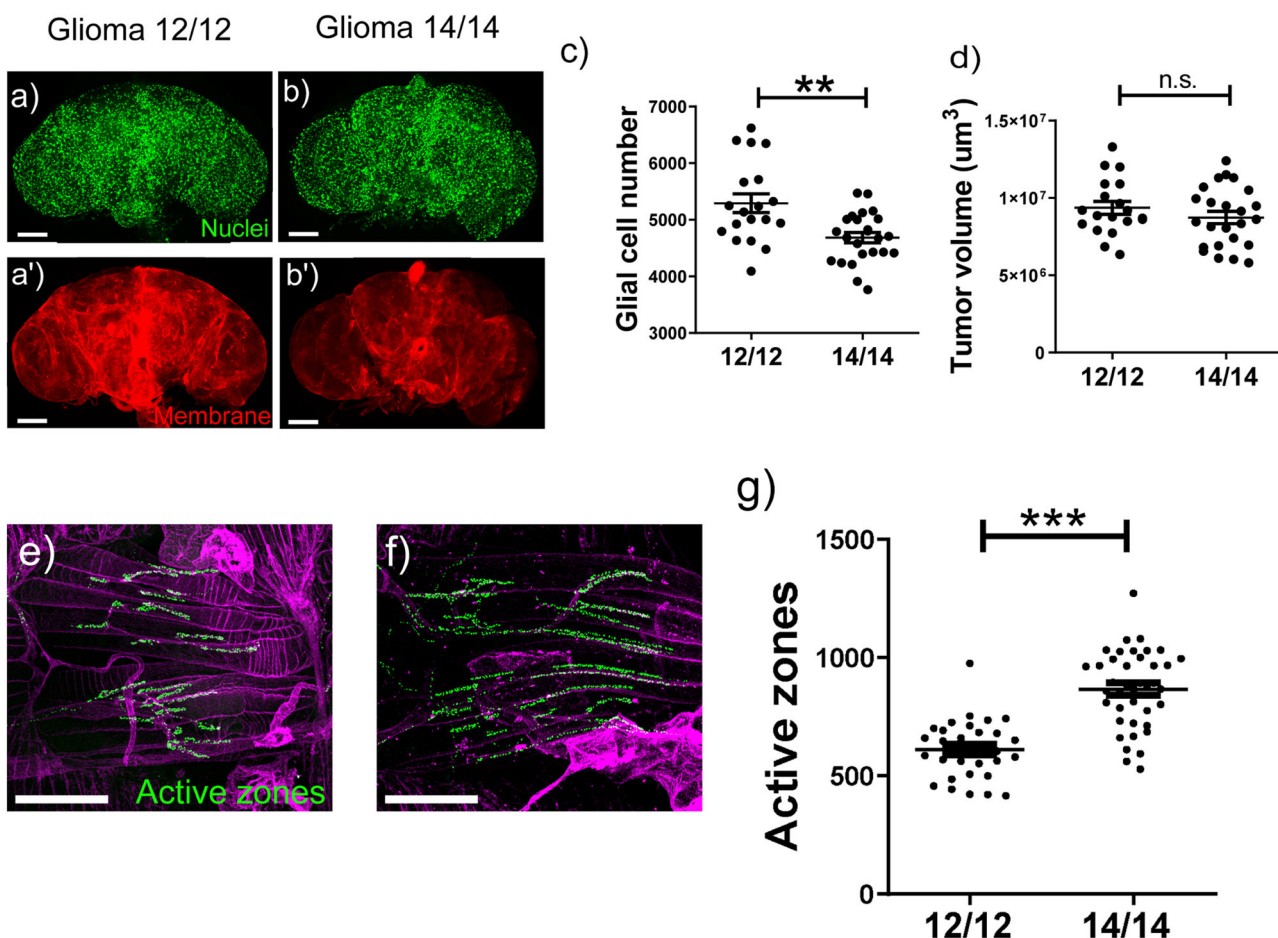

**Fig. 5 Re-adjusting environmental cues to internal rhythms after GB induction reduces tumor proliferation. a**, **b** Confocal microscopy images of the whole adult brain from control and treated glioma flies. After 4 days of tumor onset under regular 12:12 L:D conditions, animals were maintained in either 12:12 or 14:14 L:D conditions. Glial membrane is tagged in red and glial nucleus in green. **c**, **d** Quantification of glial cell number and glial membrane volume of the whole brain per animal, $N > 18$ flies (ANOVA, post-hoc Bonferroni) (scale bar, 100 μm) (**$p$-value<0,01). **e**, **f** Confocal microscopy images of adult NMJ from glioma 12:12 and 14:14. (**e**) and GT (**f**) flies. Active zones are tagged in green. **g** Quantification of active zones per animal for GC and GT, $N = 31$ flies (ANOVA, post-hoc Bonferroni) (scale bar, 50 μm) (***$p$-value < 0.001).

growth, mainly through alterations in the sympathetic nervous and hormonal systems, melatonin levels, and tumor immunity[35]. In addition, some tumors may re-entrain other peripheral clocks of metabolic tissues, thus rewiring systemic metabolic homeostasis[36]. Brain tumors are particularly responsive to autonomous circadian alterations. For instance, restoring the molecular clock in neuroblastoma acts as tumor suppressor[37]. Age of GB onset and tumor growth rate correlates with several genetic variants as well as disturbed expression of circadian pathway genes[38,39].

Despite these data, there are few examples of tumor-induced circadian alterations in the Suprachiasmatic Nucleus (SCN, the central pacemaker in mammalian brain). This is mainly due to a lack of systematic studies of circadian behavior in cancer patients,

with the exception of sleep. Indeed, most GB patients suffer from sleep disturbances and drowsiness[40] despite hypothalamus and thalamus (the main regions associated with circadian and sleep regulation) are rarely affected by GBMs. However, sleep regulation in humans is complex, with many different brain structures and pathways involved whose alteration due to a brain cancer can influence sleep, such as Hypocretin/Orexin and melatonin concentrating hormone neurons, VTA in the midbrain and dorsal raphe[41]. In addition, it cannot be ruled out that cranial radiation (often employed with primary brain tumor patients) and other treatments may also cause sleep disruptions[42]. And lastly, fatigue and sleep disturbances frequently happen together with other neuropsychological symptoms, such as neurodegeneration[41]. Our hypothesis that GB could be considered as a neurodegenerative

disease would enhance such disruptions by affecting other brain functions related to sleep quality such as neurodegenerative-induced cognitive impairment.

Intriguingly, recent work has shown that GB proliferation and progression require active glutamatergic neuron-to-glioma synapses[43,44], something that we validated and extended to intra-tumoral synapses in this *Drosophila* model (submitted manuscript). In parallel to this process, we proved that GB may be treated as a neurodegenerative disease, at least for some of the most typical features[8,13,45]. Glioma progression causes a reduction in the number of synapsis at NMJs. In this work, we also confirm that PDF neurons show neurodegeneration signals (i.e. loss of neuron-to-neuron synapses) that are reflected in alterations of circadian locomotor activity. Lowering acetylcholine levels provokes a reduction in synapse number and also makes the circadian output (likely the entrainment) weaker[22], in line with the impact of GB-induced neurodegeneration on biological rhythms.

How this can be translated to mouse models or to the human patients may be challenging, although not unlikely. Tumoral transformation occurs in all glial cells in the *Drosophila* model, whereas in human GB, only one cell or a small group of trans-formed glial cells gives rise to the tumor. Wherever the GB initiates, its rapid growth and expansion will probably induce neurodegeneration in some of the neuronal nuclei that can affect the central pacemaker[46]. The fact that rescuing neurodegeneration in PDF neurons does not rescue rhythmicity (measured by locomotor activity), but it does rescue the morning anticipation, may have different interpretations. The simplest one is that the rescue of synapse number in these neurons would be insufficient to restore every aspect of tumor-induced phenotypes such as circadian rhythm, tumor development and survival. In fact, it affects the morning anticipation, a circadian function that is directly regulated by PDF neurons, when locomotion is not compromised. A complete rescue of circadian rhythms might require to induce synaptogenesis in several circadian clusters simultaneously. Another possible explanation could be that a full rescue of biological rhythms would need a specific amount of PDF and/or other peptides necessary for the process. Further experiments are needed to confirm either of these two possibilities.

Chronotherapy has emerged as an innovative and effective therapeutic strategy. It involves not only modifying the environmental cues (re-entrainment, typically by feeding) but also adding chronomodulators or melatonin as usual treatments[47]. The final aim is to restore the molecular clock of the tumor itself. Correct sleep/awake cycles and timing for food intake delays cancer progression and acts as prognosis factor[48]. The first step would be to determine whether or not the GBM mammalian model reproduces the circadian alterations similarly to the *Drosophila* GB. This would require a detailed study of the circadian period of early diagnosed GB patients by measuring different parameters (hormonal levels, temperatures, blood pressure, sleep, among others). Depending on the stage of GB, it is expected that, for some patients, the circadian period would not be totally disrupted but lengthened or shortened. If this is the case, re-adjusting the central clock to the GB-imposed period by feeding or by light in a well-established GBM mouse model might extend the lifespan or retard the tumor growth, as it happened in the *Drosophila* GB. In any case, it seems clear that the implementation of a cycle other than 12:12 for long periods in human patients is not feasible. Nevertheless, glioma patients might benefit positively from a reinforcement of their slightly altered circadian rhythms and sleep by following certain routines.

A surprising result was that control animals in 14:14 L:D cycles lived considerably longer than their counterparts at 12:12 L:D.

Data from flies and mammals suggested that a correct circadian synchronization improve general fitness and survival[49]. However, in the case of flies all experiments were performed at 25 °C, whereas in our case we used 29 °C, a challenging temperature for *Drosophila* fitness. Indeed, both circadian and sleep patterns are quite different at such high temperature compared to the ones at 25 °C[50,51], which may account for the increased lifespan we see in our experiments at 29 °C under a 14:14 L:D regime.

But the most intriguing question is: why re-adjusting (length-ening) the daily period to the internal period imposed by the GB is sufficient to slow down tumor development? It is tempting to speculate that neurons may have a circadian activity coordinated with the circadian central clock that would improve the general neuronal fitness, thus impairing tumor progression. Fitness has an elusive definition, but it had been related to cell competition[52,53]. In this process, loser cells (wt neurons) might be actively eliminated by winners (GB cells). If this would be the case, some of the well-described cell competition genes should play a role, showing differences in expression or function in GB brains under different L:D regimes. However, the possible link between circadian rhythms and cell competition has not been described so far. Alternatively, fitness might also be related to a circadian reprogramming-like state, in which a new transcriptional program adjusted to other-than-24h period is established. In a regular 24 h day time, the lack of coordination between the internal period and external cues may cause a slightly altered transcriptional program. Under healthy conditions, this lack of coordination does not cause any apparent phenotype, but when neurons are confronted with a GB, their decreased "circadian fitness" makes them more susceptible to tumor effects. Further studies are needed to validate either of two hypotheses, although we do not rule out a third one that may render a more convincing explanation. However, novel conceptual ideas reflected in this manuscript might be of special relevance to augment the life quality and lifespan of GB patients, using minimal invasive treatments that could improve the efficacy of more conventional approaches.

## Materials and methods

**Fly stocks and genetics**. All fly stocks were maintained at 25 °C (unless otherwise specified) on a 12/12 h light/dark cycles at constant humidity in standard medium. The stocks used from Bloomington Stock Center were: *tub-Gal80ts* (BL-7019), *Repo-Gal4* (BL-7415), *UAS-myr-RFP* (BL-7119) *UAS-LacZ* (BL-8529), *Elav-LexA* (BL52676), *pdf-LexA* (a gift from F. Rouyer), *UAS-dEGFRλ; UAS-dp110CAAX* (gift from R. Read, Read et al., 2009), LexAop-Rheb (gift from Nuria Romero), UAS-PI3K and UAS-GSK3 (gift from J. Botas).

The glioma-inducing line contains the *UAS-dEGFRλ,UAS-dp110CAAX* transgenes that encodes for the constitutively active forms of the human orthologues PI3K and EGFR respectively[2]. *Repo-Gal4* line drives the *Gal4* expression to virtually all glial cells and precursors[54,55] and combined with *UAS-dEGFRλ*, *UAS-dp110CAAX* line allow us to generate a glioma composed by the entire population of glial cells[56]. *Elav-LexA* and *pdf-LexA* lines are expressed in virtually all neurons or restricted to pdf-expressing neurons, allowing us to manipulate neurons in a glioma combining *LexA* and *Gal4* expression systems[57].

Tub-Gal80[TS] is a repressor of the Gal4 activity at 18 °C, though at 29 °C is inactivated[58]. The *tub-Gal80ts* construct was used in all the crosses to avoid the lethality caused by the glioma development during the larval stage. The crosses were kept at 17 °C until the adult flies emerged. To inactivate the Gal80ts protein and activate the Gal4/UAS system to allow the expression of our genes of interest, the adult flies were maintained at 29 °C for 7 days except in the survival assay (flies were at 29 °C until death).

Due to the inherent variability of tumor growth and the use of different reporters, we used the appropriate control and glioma genotypes that include them and performed the experiment in parallel for each grouped panel (at least three times). To avoid circadian and aging variations, all samples for immunostaining, qPCR and NMJ quantification were collected in animals at 7 days post-GB induction and at the same subjective time, between 10 and 12AM (ZT1-3).

**Immunostaining and Image acquisition**. Adult brains were dissected and fixed with 4% formaldehyde in phosphate-buffered saline for 20 min whereas adult NMJ were fixed for 10 min; in both cases, samples were washed 3 × 15 min with

PBS + 0.4% triton, blocked for 1 h with PBS + 0.4% triton+BSA 5%, incubated overnight with primary antibodies, washed 3 × 15 min, incubated with secondary antibodies for 2 h and mounted in Vectashield mounting medium, with DAPI in the case of the brains. The primary antibodies used were mouse anti-repo (1:200, DSHB) to recognize glial nuclei, mouse anti-bruchpilot (NC82- 1:50, DSHB) to recognize the presynaptic protein Bruchpilot, rabbit anti-HRP (1/400, Cell Signalling) to recognize membranes, mouse anti-pdf (DHSB, 1:2000), rabbit anti-GFP (1:500, DSHB) and mouse anti-Elav (1:100, DSHB) to recognize neuron nuclei. The secondary antibodies used were Alexa-488 or 647 (1:500, Life Technologies). Images were taken by a Leica SP5 confocal microscopy.

**Locomotor activity assays.** Crosses were kept at 17 °C to avoid glioma development during larval and pupal stages. 32 1-to-4 day-old male flies from the adequate genotypes were collected and placed individually in small glass tubes containing standard food at 29 °C and 70% of humidity. They were monitored for locomotor activity using the *Drosophila* Activity Monitor System (Trikinetics, Waltham, MA, USA). Flies were kept in light-dark (12 h/12 h) conditions one day for entrainment, and then maintained at L:D or shifted to constant darkness (D:D) for 5 days depending on the experiment. Assays are divided into three temporal segments that recapitulate the entire GB development process to avoid the dehydration that flies experiment after 6 days: early GB flies have just eclosed (0–5-day old); medium GB flies are cultured at 29 °C for 5 days before introducing them in glass tubes; advanced GB flies stay 10 days at 29 °C before testing in DAMs. For the D:D experiments, flies are kept in L:D conditions until the beginning of the experiments.

Actograms are double-plotted graphs representing absolute activity levels of one particular fly that are representative of the movement of a given genotype. All analyses were done on days 1–5. Actogram graphs were produced by means of the set of R-packages defined by Rethomics (Geissmann et al, 2019). Data analysis was done with FaasX software 1.31, a gift from F. Rouyer, which is derived from the Brandeis Rhythm Package.

For D:D data, rhythmic flies were defined by x2 periodogram analysis of a 5d dataset with the following criteria (filter ON): power >40, width >1.5 h, with no selection on period value. The FaasX 1.31 software arbitrarily assigned these thresholds by default. We also analyzed our data using more conventional criteria (as defined by Dr Rouyer´s lab; power >20, width >2 h), showing similar results. Power is the height of the periodogram peak and give the significance of the calculated period. For period calculation, only data from rhythmic flies were considered. For average power calculation, all data were used. When FaasX software calculates a period below 14 h, we doubled it to adjust to a 24 h period. Movement analysis (Fig. 3f) used the set of R-packages defined by Rethomics (Geissmann et al, 2019). The solid lines and the shaded areas show population means and their 95% bootstrap confidence interval, respectively. Experiments were reproduced at least two times with very similar results.

**RNA extraction, reverse transcription and qPCR.** For RNA extraction, 1- to 4-day-old male adults were entrained to a 12:12 h L:D cycle for 7 days at 29 °C and then collected on dry ice at ZT 6. Total RNA was extracted from 30 heads of adult males of the Control (*repo > LacZ*) and Glioma (*repo > UAS-dEGFR^□^, UAS-dp110^CAAX^*) genotypes after 7 days of glioma development. RNA was extracted with TRIzol and phenol chloroform. Total RNA concentration was measured by using NanoDrop ND-1000. cDNA was synthetized from 1 mg of total RNA using M-MLV RT (Invitrogen). cDNA samples from 1:5 dilutions were used for real-time PCR reactions. Transcription levels were determined in a 14 mL volume in duplicate using SYBR Green (Applied Biosystem, Foster City, CA) and 7500qPCR (Thermo Fisher, Waltham, MA). We analyzed transcription levels of *clk, cyc, tim, per, pdf, cry* and *Rp49* as housekeeping gene reference.

Sequences of primers were: RP49 F: GCATACAGGCCCAAGATCGT, R: AACCGATGTTGGGCATCAGA; clk (Sigma) F: ATGACTCGGATTCAACG TCCAT, R: CGTTGGCTTTGCGAACTGA; cyc (Sigma) F: CCGAAGGACATA GGCAAGGTT, R: CGGTCTTAACGGGCAACATG; tim (Sigma), F: TGGGAA GCGGACTATGAACTG, R: TGGCGACTTCGGAGGAGTAT; per (Sigma) F:CCAATGGCACCAACATGCT, R: TCGTCGGGAACCTTGTAGCT; pdf (Sigma) F: CGCTATGTGCGCAAGGAGTA, R: TGGCCGGGACTGAACTGT.

After completing each real-time PCR run, outlier data were analyzed using 7500 software (Applied Biosystems). Ct values of duplicates from three biological samples were analyzed calculating 2D:DCt and comparing the results using a *t* test with GraphPad (GraphPad Software, La Jolla, CA).

**Viability and survival assays.** Lifespan was determined under 12:12 h L:D cycles at 29 °C conditions. Three replicates of 30 1- to 4-day-old male adults were collected in vials containing standard *Drosophila* media and transferred every 2–3 days to fresh *Drosophila* media.

**Quantification.** Glial network was marked by a membrane-targeted myristoilated RFP UAS-transgene (*UAS-myr-RFP* reporter) specifically expressed under the control of repo-Gal4. The total volume was quantified using Imaris surface tool (Imaris 6.3.1 software). Glial nuclei were marked by staining with the anti-Repo (DSHB). The number of Repo+ cells and number of synapses (anti-nc82, DSHB)

were quantified by using the spots tool in Imaris 6.3.1 software. We selected a minimum size and threshold for the spot in the control samples of each experiment: 0.5 μm for active zones and 2 μm for glial cell nuclei. Then we applied the same conditions to the analysis the corresponding experimental sample.

**Statistics and reproducibility.** Graphs shown in Figs. 1k, 2d, g, h, m, 3d, 4e, f, k, 5c, d, g are dot plots depicting the median and the SEM.

The results were analyzed using the GraphPad Prism 5 software (www.graphpad.com). Quantitative parameters were divided into parametric and non-parametric using the D'Agostino and Pearson omnibus normality test and the variances were analyzed with F test. Student's *t* test and ANOVA test with Bonferroni's post-hoc were used in parametric parameters, using Welch's correction when necessary. To the non-parametric parameters, Mann–Whitney test and Kruskal–Wallis test with Dunns post-hoc were used. The survival assays were analyzed with Mantel–Cox test. The p limit value for rejecting the null hypothesis and considering the differences between cases as statistically significant was $p < 0.05$ (*). Others p values are indicated as ** when $p < 0.01$ and *** when $p < 0.001$.

**Reporting summary.** Further information on research design is available in the Nature Research Reporting Summary linked to this article.

## Data availability

All material requests and correspondence should be addressed to sergio.casas@isciii.es and famartin@cajal.csic.es. The source data behind the graphs in the paper are depicted as supplementary data. Raw data and detailed description of data analysis will be available on reasonable request to corresponding authors.

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

## Acknowledgements

The authors thank E. Beckwith, A. Ferrús, M. Milán, and members of the Cajal Fly hub for comments on the manuscript. We also appreciate flies and reagents from F. Rouyer, N. Romero, J. Botas, R. Read, and the Bloomington and VDRC stock centers. We would like to thank the FlyBase for its wealth of information. We acknowledge the support of the Confocal Microscopy unit and Molecular Biology unit at the Cajal Institute. We want to thank the continuous support from Emirates Khalifa Capital to this project and to P.J. This work was supported by the Spanish Research Agency (MICINN) under the grant PGC2018-094630-B-I00 to S.C.T. and grant PGC2018-094630-B-100 to F.A.M., cofinanced by the European Regional Development Fund (ERDF). F.A.M. is a recipient of a Ramon y Cajal contract (2014-14961). C.G.B. is a recipient of a FPU predoctoral fellowship (FPU19/04449 -MEFP-).

## Author contributions

Conceptualization: P.J., S.C.T., F.A.M.; Methodology: P.J., S.C.T., F.A.M., C.deP.; Validation: P.J., S.C.T., F.A.M.; Formal analysis: P.J., S.C.T., F.A.M., C.deP., C.G.B.; Investigation: P.J., S.C.T., F.A.M., C.deP., C.G.B.; Writing—original draft: S.C.T., F.A.M.; Writing—review and editing: P.J., S.C.T., F.A.M., C.deP., C.G.B.; Supervision: S.C.T., F.A.M.; Funding acquisition: P.J., S.C.T., F.A.M.

## Competing interests

The authors declare no competing interests.
