## [Peer Review File · Communications Biology]

Reviewers' comments:

Reviewer #1 (Remarks to the Author):

Alignment between glioblastoma internal clock and environmental cues ameliorates survival in *Drosophila*.

Summary: The authors describe a *Drosophila* model of glioblastoma(GB)-induced neurodegeneration which potentially disrupts global circadian rhythmicity via decreased insulin signaling and synapse number. They demonstrate that rhythmicity can be rescued in GB-bearing flies by increased insulin signaling in all neurons (via *rheb*) and show that insulin signaling does alter the number of active zone synapses of PDF-expressing neurons. While insulin signaling (via *rheb*) in PDF-expressing neurons could not rescue period, power, or survival in glioma-bearing flies, it could rescue morning anticipation in early glioma-bearing flies. Notably, GB-bearing flies appear to demonstrate a lengthened period before becoming arrhythmic. Authors suggest that re-aligning environmental light cues to altered internal rhythms impairs tumor growth and offers therapeutic benefit.

Major Comments: While this work is quite intriguing in theory, the applicability of this GB-model to a microenvironmental question such as the interaction of the cancer cells to circadian circuitry is difficult to parse out. Numerous studies in mammalian and human populations stress the importance of the microenvironment to glioma initiation, progression and inevitable survivability. The lack of a proper microenvironmental context makes these data difficult to interpret and extrapolate to its actual biological relevance. For example, numerous studies from the Monje lab at Stanford University, the Winkler lab at University Hospital Heidelberg, and from Shawn Hervey-Jumper at UCSF, have highlighted that GBM synaptically integrate into the neural circuitry of the brain, that synaptic integration and synaptic connections increase GBM progression, while work from the Gutmann lab at WashU and others highlights the importance of interactions with other glial subtypes in glioma progression (such as microglia). These components are lacking in this model and in many ways are opposite findings to those in mammalian models as the current work suggests that increases in synapse may be beneficial to survival. At a minimum, this work needs to be discussed in the context of the aforementioned findings. Specific comments are below.

Major Concerns:

- How frequently do mutations in EGFR and PIK3CA drive human GBs? How frequently are these passenger mutations? This information is imperative to relevance of this model.
- Do human GBs develop in the same locations as fly GBs? There are no indications to the spatial component of the GBs in the fly brain in this manuscript. In humans, GBMs rarely occur in the hypothalamus and thalamus (the main regions associated with circadian and sleep processes), thus it is hard to assess if the physical proximity of the tumors is necessary for these findings to be recapitulated in mammalian populations (ie humans). If so, the applicability to human GBM is limited. More information on the size, scope, and location of these tumors is needed. This is important as the decrease in synapse number could be attributed to the tumor physically disrupting synapses or GB-induced neurotoxicity. Clarity on exactly where the 'active zones' are located in relation to the tumor is needed.
- As written, it appears that every glial cells has the mutations. Does this mean there are no non-pathological glial cells in these animals. If so, how does that physiologically not disrupt brain functioning and how would that relate to real GBMs in humans in which the putative cell of origin is believed to be the oligodendrocyte precursor cell, or at a minimum are not believed to arise from all glial cells. If not, clarity on how normative glial cells integrate into this model should be provided. What are other known causes of GB-induced neurodegeneration (other than decreased neuronal insulin signaling), that may contribute neurodegeneration? Are glial cells in the microenvironment adversely affected? In other mammalian models and humans, deficits in microglia, astrocytes, oligodendrocytes/myelin, and the blood-brain barrier all contribute to neurodegeneration.
- The number of animals used for different treatment conditions is not always specified. In general, how are statistics performed? For example, in figure 4e,f, the authors should be running statistics on a PER ANIMAL basis, not on a per count basis (ie N=813). If sections from 6 flies were assessed, counts should be compared per animal so n=6. This may change the significance of

findings dramatically. This is true for analyses in virtually all figures.

Specific Concerns:

- How was 1-5 days determined to represent "early glioma" versus 5-10 days as "medium glioma"? How does this correspond to human disease progression?
- Why was a power of 40 chosen as the cut off between rhythmic versus arrhythmic? Is this arbitrary based on your model.
- Only a minority of animals are arrhythmic in early GB – is there any biological explanation for this variance?
- Figure 1A versus 1E controls seems to have different period lengths. Is this true?
- Figures 1B and 1F are labeled as rhythmic but appear quite arrhythmic, especially compared to controls – significance values on these graphs would be helpful
- MISSING CONTROL: Figure 1 should have an actogram and power values measuring the periodicity of non-GB-bearing flies with the ELAV>rheb rescue
- Why did authors choose to quantify synapse number at 7 days old and 7 days post-tumor induction for Figure 2? Why not 10 when the phenotype would likely be more pronounced?
- It is unclear whether the decrease in PDF positive puncta is due to overall loss of neurons or the same number of neurons expressing less pdf-? Both? That other clock genes are by and large unaffected would suggest GB specifically affects PDF.
- MISSING CONTROL: Why were the number of PDF punctae not quantified in control>rheb flies? If insulin signaling increases active zone number, it may also increase pdf punctae at base-line without glioma given, "the inter-dependence between the number of active zones and PDF protein levels: modifying either one affects the other" -Line 159-160. This could imply that insulin signaling is contributes, but is not sufficient, to rescue pdf expression/rhythmicity; this is supported by PDF-rescued GB-bearing flies having similar period, power, and survival to GB-bearing flies
- For Figures 4 and 5, are all glial cells in the membrane part of the tumor volume? Where are the tumor borders?
- It is unclear which data points the significance values apply to throughout the entirety of the manuscript.
- The N values throughout the manuscript are highly variable in between experiments and different treatment conditions.
- While it may be intriguing to extend these findings to humans glioma patients; implementation of a 14/14 LD cycle seems neither feasible nor accessible.
- Line 28 – Should read "glial cells", not "glial cells"
- Line 68 – Should read "BMAL1", not "BMLA1"
- Figure 1A-1H – axis labels are unclear; how many animals are represented in each graph? In general, figure quality is quite low and difficult to read
- Figure 1I – Graph lacks a title and clear axis labels/units, unclear which data values are significant
- Figure 1I does not have a figure legend
- Figure 2F legend does not clearly state the age of flies when active zones were quantified
- Figure 2L is confusing, how is that the insulin signaling rescue has no significant difference in PDF punctae number with the control and no significant difference in PDF punctae number with the glioma-bearing flies?
- Figure 3D – What day into glioma progression are these flies? The
- Figure 3F lacks any p-values/significance and lacks a Y-axis label
- Figures 3G-3I are not legible
- Figure 4 and 5 titles are inaccurate – 14/14 regime improves survival but there is no evidence that the tumor progression is impaired: only glial cell number is increased by LD 12/12, and tumor volume is the same between 12/12 and 14/14.
- In Figure 4, how are specifically glial membranes and nuclei labeled?
- In Figure 4F, why is there any tumor volume for controls?
- In Figures 4 and 5, it would be clearer if labels were 12/12 rather than LD as the 14/14 also experience light/dark cycles.
- (p)50 is not explained in the figure legend
- Figure 5D Y-axis has strange numbering/labeling

Reviewer #2 (Remarks to the Author):

In this manuscript by Jarabo et al., the authors look at the interaction between glioblastoma-induced neurodegeneration and circadian rhythms in flies. The authors find that glioblastoma normally disrupts circadian rhythms in a majority of animals due to an increased prevalence of neurodegeneration. Rescuing neurodegeneration in all neurons rescues disrupted rhythms. While this may be due to changes to pdf-expressing neurons (the authors find glioma-induced changes in active zones and pdf expression), they surprisingly find that rescuing neurodegeneration in only pdf neurons has no effect on rhythmicity as a whole. However, pdf rescue does seem to rescue a glioma-dependent decrease in morning anticipation. Finally, and most interestingly, the authors find that 1) flies with glioma that are still rhythmic have lengthened periods and, 2) lengthening day length to a 14:14 LD cycle delays tumor progression and increases survival. Overall, the manuscript is novel and the statistics seem to be appropriate. The manuscript will be of interest to cancer biologists, chronobiologists, and developmental biologists. The manuscript will be improved when the authors address the following concerns:

Fig 1.

1A-H: It would be helpful if the actograms were shaded or filled in so that the reader can more easily determine rhythmicity/arrhythmicity in each trace.

1I: Please define the X axis (days of tumor development?) and A. Please define the red line.

1I: Please make it more clear what groups are significantly different from one another. It's difficult to tell with the authors' current layout.

Table 1: Was the period increase in glioma flies statistically significant?

Fig 2.

2H. The authors should show individual data points if possible

2H. The authors extracted RNA at ZT 6 -- was pdf constitutively low, or just at this time point?

Fig 3.

3G-I: The figure is very pixelated and may need to have increased resolution for readability

3G-I: Have you performed statistics to show that there is indeed a difference and that glioma + pdf insulin signaling is able to restore the morning anticipation peak?

Fig. 4

Why does a lengthened day decrease the total glial cell number in glioma flies to that of control flies, but not tumor volume?

4K: Are CLD vs. C14 and GLD vs. G14 significantly different based on post hoc tests? Instead of increasing the number of active zones in the G14 condition a lengthened day could ostensibly be reducing the number of active zones in the C14 flies.

The authors should comment on whether a 14:14 LD cycle increasing life span in wild-type flies is consistent with previous literature.

General:

Are glioma cells in flies rhythmic and, if so, do they show similar circadian periods to non-cancerous cells?

There are several grammatical/spelling errors throughout the manuscript. Please proofread it carefully prior to the next submission.

Reviewer #1 (Remarks to the Author):

Alignment between glioblastoma internal clock and environmental cues ameliorates survival in *Drosophila*.

Summary: The authors describe a *Drosophila* model of glioblastoma (GB)-induced neurodegeneration which potentially disrupts global circadian rhythmicity via decreased insulin signaling and synapse number. They demonstrate that rhythmicity can be rescued in GB-bearing flies by increased insulin signaling in all neurons (via *rheb*) and show that insulin signaling does alter the number of active zone synapses of PDF-expressing neurons. While insulin signaling (via *rheb*) in PDF-expressing neurons could not rescue period, power, or survival in glioma-bearing flies, it could rescue morning anticipation in early glioma-bearing flies. Notably, GB-bearing flies appear to demonstrate a lengthened period before becoming arrhythmic. Authors suggest that re-aligning environmental light cues to altered internal rhythms impairs tumor growth and offers therapeutic benefit.

1) Major Comments: While this work is quite intriguing in theory, the applicability of this GB-model to a microenvironmental question such as the interaction of the cancer cells to circadian circuitry is difficult to parse out. Numerous studies in mammalian and human populations stress the importance of the microenvironment to glioma initiation, progression and inevitable survivability. The lack of a proper microenvironmental context makes these data difficult to interpret and extrapolate to its actual biological relevance. For example, numerous studies from the Monje lab at Stanford University, the Winkler lab at University Hospital Heidelberg, and from Shawn Hervey-Jumper at UCSF, have highlighted that GBM synaptically integrate into the neural circuitry of the brain, that synaptic integration and synaptic connections increase GBM progression, while work from the Gutmann lab at WashU and others highlights the importance of interactions with other glial subtypes in glioma progression (such as microglia). These components are lacking in this model and in many ways are opposite findings to those in mammalian models as the current work suggests that increases in synapse may be beneficial to survival. At a minimum, this work needs to be discussed in the context of the aforementioned findings. Specific comments are below.

We want to thank the reviewer for bringing these issues up. In fact, we fully agree with the relevance of neuron-glia synapses as well as with the great relevance of micro-environment for GB progression.

We have been working in the relation of GB cells with neighboring neurons for more than a decade. We found that *Drosophila* GB cells also reproduced Tumor Microtubules (TMs) described by Winkler's group in 2015, and indeed we tightly collaborated with his group to describe the role of TMs and their links with WNT, JNK and MMP signaling pathways (<https://pubmed.ncbi.nlm.nih.gov/31846454/>). In this work we concluded that *Drosophila* GB model reproduced the main cell-to-cell communication features described for human GB cells. Thus, under the well-known limitations of every animal model, we believe that the results obtained in *Drosophila* in this context are comparable to those obtained in mammalian models and human cells.

The reviewer mentions the works by Monje, Winkler and Shawn Hervey-Jumper labs regarding the contribution of glutamatergic synapses to GB progression. Actually, we have reproduced these results in our *Drosophila* glioma model and the manuscript is currently under review (please see <https://www.biorxiv.org/content/10.1101/2021.10.14.464400v1.full>). Our findings were in the same line, thus suggesting that synaptic contacts between neurons (pre-synaptic component) and GB cells (post-synaptic) is another conserved feature between *Drosophila* and mammals. Actually, we extended the idea that intra-tumoral synapsis (i.e. both pre and post-synaptic components are GB cells) were also necessary for *Drosophila* GB development. To summarize this work, GB cells receive synaptic contacts from neurons and other GB cells, and such synaptic processes stimulate GB progression.

However, these concepts are distinct from the one discussed in this manuscript and we have clarified this in the text. In this paper we claim that GB progression provokes a reduction in the number of synapses of neighboring healthy neurons (presynaptic) with other neuronal cells (post-synaptic). We have demonstrated that GB cells causes synapse loss and neuronal decay.

Therefore, we proposed that GB causes neurodegeneration. We modified the text to clarify this point (from now on, added text is underlined, line 403):

"Intriguingly, recent work has shown that GB proliferation and progression require active glutamatergic neuron-to-glioma synapses 38,39, something that we validated and extended to intra-tumoral synapses in this *Drosophila* model (submitted manuscript). In parallel to this process, we proved that GB may be treated as a neurodegenerative disease, at least for some of the most typical features 8,13,40. Glioma progression causes a reduction in the number of synapses at NMJs. In this work we also confirm that PDF neurons show neurodegeneration signals (i.e. loss of neuron-to-neuron synapses) that are reflected in alterations of circadian locomotor activity."

Major Concerns:

2) • How frequently do mutations in EGFR and PIK3CA drive human GBs? How frequently are these passenger mutations? This information is imperative to relevance of this model.

In this manuscript we use a model previously established by Read and cols (<https://www.ncbi.nlm.nih.gov/pmc/articles/PMC2636203/>), where authors described the model as follows:

"... the most frequent genetic lesions in gliomas include mutation or amplification of the Epidermal Growth Factor Receptor (EGFR) tyrosine kinase. Glioma-associated EGFR mutant forms show constitutive kinase activity that chronically stimulates Ras signaling to drive cellular proliferation and migration [1],[2]. Other common genetic lesions include loss of the lipid phosphatase PTEN, which antagonizes the phosphatidylinositol-3 kinase (PI3K) signaling pathway, and activating mutations in PIK3CA, which encodes the p110 α catalytic subunit of PI3K [1],[2]."

Besides, there are numerous publications describing the high frequency of EGFR and PI3K pathways in GB cells. However, these studies show numerical differences in the contribution of diverse mutations in genes affecting these pathways. According to the Collection of Somatic Mutations in Cancer database (see COSMIC), mutations in PI3K and EGFR pathways are among the most frequent ones found in GB patients. This is now described in the introduction as follows (line 35): "the Collection of Somatic Mutations in Cancer database (COSMIC) indicates that mutations in PI3K and *egfr* genes are among the 20 most frequent mutated genes found in GB patients (i.e. Astrocytoma grade IV), showing a corresponding frequency of 9% and 14%. Other components of PI3K and EGFR pathways are also commonly mutated, such as *pten* (22%) and *NF1* (11%), respectively".

3) • Do human GBs develop in the same locations as fly GBs? There are no indications to the spatial component of the GBs in the fly brain in this manuscript. In humans, GBMs rarely occur in the hypothalamus and thalamus (the main regions associated with circadian and sleep processes), thus it is hard to assess if the physical proximity of the tumors is necessary for these findings to be recapitulated in mammalian populations (ie humans). If so, the applicability to human GBM is limited. More information on the size, scope, and location of these tumors is needed. This is important as the decrease in synapse number could be attributed to the tumor physically disrupting synapses or GB-induced neurotoxicity. Clarity on exactly where the 'active zones' are located in relation to the tumor is needed.

In contrast to the majority of brain tumors, Glioblastoma Multiforme (GB) infiltrates extensively (for instance, see <https://www.ncbi.nlm.nih.gov/pmc/articles/PMC2039798/>). Despite surgical removal of the GB tissue and the supramaximal resection approach, complete removal is not possible because of where and how these tumors infiltrate brain tissue (<https://pubmed.ncbi.nlm.nih.gov/32027343/>) (<https://pubmed.ncbi.nlm.nih.gov/34123831/>). In consequence, GB recurrence is very high (up to 90%), the median overall survival is 15-18 months in clinical trial populations and less than 10% of patients survives beyond 5 years. Therefore, currently there is not a curative treatment for diffuse gliomas. Our *Drosophila* GB model reproduces faithfully the diffuse tumor, as virtually all glial cells are transformed into tumoral cells by means of the *repoGal4* driver (<https://www.ncbi.nlm.nih.gov/pmc/articles/PMC2636203/>;

<https://pubmed.ncbi.nlm.nih.gov/28405401/>). This means that all active zones are surrounded by tumoral glial cells. To avoid this confusion, we have included further details regarding the distribution of glioma cells in the *Drosophila* model, as follows (line 32): "this model is based on the genetically driven constitutive activation of two signaling pathways (i.e. the activated forms of Epithelial Growth Factor Receptor -EGFR- and PI3K) in all glial cells, thus transforming the whole glial population".

In the case of brain tumor patients, sleep disturbances are frequently reported along with fatigue (<https://pubmed.ncbi.nlm.nih.gov/28340256/>). Indeed, most GB patients suffer from depression, fatigue, sleep disturbances, drowsiness and cognitive impairment (<https://sigmapubs.onlinelibrary.wiley.com/doi/10.1111/j.1547-5069.2007.00144.x>) (<https://pubmed.ncbi.nlm.nih.gov/27286798/>). In humans, sleep regulation is complex with many different brain structures and pathways involved (<https://pubmed.ncbi.nlm.nih.gov/28231463/>). We agree with the reviewer that hypothalamus and thalamus are rarely affected by GB, but sleep in brain cancer patients may be affected by the alterations of other circuits, such as Hypocretin/Orexin and melatonin concentrating hormone neurons, VTA in the midbrain and dorsal raphe, among many others (see <https://www.ncbi.nlm.nih.gov/pmc/articles/PMC6600154/>). Also, immune alterations that are deregulated by cancer can influence sleep, although this is unlikely in the case of GB. In addition, it cannot be ruled out that cranial radiation (often employed with primary brain tumor patients) and other treatments may also cause sleep disruptions (<https://pubmed.ncbi.nlm.nih.gov/27286798/>). And lastly, "*fatigue and sleep disturbances frequently occur along with other neuropsychological symptoms including depression and cognitive impairment, which may either contribute to or be the result of ongoing sleep disruption*" (<https://www.ncbi.nlm.nih.gov/pmc/articles/PMC6600154/>). Our hypothesis that GB could be considered as a neurodegenerative disease would enhance such disruptions by affecting other brain functions related to sleep quality such as neurodegenerative-induced cognitive impairment. In fact, the impact of distance between GB cells and specific brain regions on disease development is a very interesting proposal, and we have the intention to pursue this in further studies. We have added all these concepts in the revised discussion of the new manuscript, adding new references (line 390):

"indeed, most GB patients suffer from sleep disturbances and drowsiness despite hypothalamus and thalamus (the main regions associated with circadian and sleep regulation) are rarely affected by GBMs. However, sleep regulation in humans is complex, with many different brain structures and pathways involved whose alteration due to a brain cancer can influence sleep, such as Hypocretin/Orexin and melatonin concentrating hormone neurons, VTA in the midbrain and dorsal raphe. In addition, it cannot be ruled out that cranial radiation (often employed with primary brain tumor patients) and other treatments may also cause sleep disruptions. And lastly, fatigue and sleep disturbances frequently happen together with other neuropsychological symptoms, such as neurodegeneration. Our hypothesis that GB could be considered as a neurodegenerative disease would enhance such disruptions by affecting other brain functions related to sleep quality such as neurodegenerative-induced cognitive impairment."

In summary, sleep used as the main output of circadian rhythms may not be the best choice, given the high number of inputs and circuits that can affect it. This is why in the discussion we propose to study circadian rhythms of individual GB patients by measuring hormonal levels, temperature or blood pressure, in addition to sleep patterns.

Finally, we would like to point out that we agree with the reviewer: the applicability of these results to the treatment of patients is hard to see. However, we believe that it is necessary to use novel strategies and approaches to improve the clinical outcome of these tumors.

4) • As written, it appears that every glial cell has the mutations. Does this mean there are no non-pathological glial cells in these animals. If so, how does that physiologically not disrupt brain functioning and how would that relate to real GBMs in humans in which the putative cell of origin is believed to be the oligodendrocyte precursor cell, or at a minimum are not believed to arise from all glial cells. If not, clarity on how normative glial cells integrate into this model should be provided.

The reviewer is correct, all glial cells are transformed into GB cells as described in the Introduction (line 32) and Materials and Methods section (lines 488 and 490). This model in *Drosophila* has contributed to recent discoveries in the patho-metabolomic pathways in GB (<https://pubmed.ncbi.nlm.nih.gov/30357574/>) and the role of YAP/TAZ contribution to the therapeutic vulnerability of Veteporfin (<https://pubmed.ncbi.nlm.nih.gov/33172899/>), among others. Indeed, our group described the role of vesicular trafficking in *Drosophila* GB and in human GB xenografts (<https://pubmed.ncbi.nlm.nih.gov/30506943/>). Therefore, our model can be considered a suitable system for human glioblastomas (see <https://pubmed.ncbi.nlm.nih.gov/31520357/>). However, despite *Drosophila* GB reproduces the type of diffuse tumor found in well-developed glioblastomas, it cannot reproduce the initial steps of GB. We have included details of the tumoral transformation of all glial cells and discussed the limitations of this system and added new references in a revised version of the discussion (line 359): "the current *Drosophila* model of GB cannot reproduce the initial steps of GBM, given that all glial cells are transformed, so the interaction between healthy and tumoral glial cells cannot be assessed. However, it reproduces faithfully the diffuse tumor phase found in well-developed human glioblastomas. For instance, it was validated regarding many of advanced human disease features, such as neurodegeneration, signaling pathways involved (like vesicular trafficking, YAP/TAZ and metabolic genes) and eventual lethality".

Besides, we aimed to distinguish whether the circadian phenotype was caused by the lack of glial function or emanated from the active degeneration of neurons. The results showed that overexpression of insulin pathway such as *rheb* in neurons exposed to GB, caused a protection on the synapses and prevented circadian disruption. These results suggest that the degeneration of neurons is responsible of the circadian phenotype.

5) What are other known causes of GB-induced neurodegeneration (other than decreased neuronal insulin signaling), that may contribute neurodegeneration? Are glial cells in the microenvironment adversely affected? In other mammalian models and humans, deficits in microglia, astrocytes, oligodendrocytes/myelin, and the blood-brain barrier all contribute to neurodegeneration.

This is an interesting point raised by the reviewer. We have previously described other causes of GB-induced neurodegeneration such as the depletion of neuronal-WNT by GB cells, JNK signaling, the production of MMPs and the expansion or tumor microtubules by GB cells (<https://pubmed.ncbi.nlm.nih.gov/31846454/>) (<https://pubmed.ncbi.nlm.nih.gov/32878880/>). Lim and cols described that soluble CD44 secreted from glioblastoma cells induces neuronal degeneration through the activation of tau pathology in the brain (<https://pubmed.ncbi.nlm.nih.gov/29622771/>). Other authors included Glioblastoma Multiforme among age-associated neurodegenerative diseases (<https://pubmed.ncbi.nlm.nih.gov/33378554/>) and described neurodegeneration in glioblastoma patients (<https://www.ncbi.nlm.nih.gov/pmc/articles/PMC4598438/>) (<https://pubmed.ncbi.nlm.nih.gov/18469825/>).

The neurodegeneration and brain tumor microenvironment in the context of gliomas is described elsewhere (<https://pubmed.ncbi.nlm.nih.gov/26411769/>)

The contribution of other glial cells cannot be addressed in this animal model, as the tumoral transformation occurs in all glial cells. The contribution of astrocytes to GB progression decreases the sensitivity of GB cells to temozolomide (<https://www.ncbi.nlm.nih.gov/pmc/articles/PMC7583902/>) and GB cells can actually activate astrocytes, namely, the tumor associated astrocytes (TAAs), to promote GB invasion in the healthy tissue (<https://pubmed.ncbi.nlm.nih.gov/30240060/>) (<https://pubmed.ncbi.nlm.nih.gov/31186414/>).

Finally, the role of the BBB to tumor progression and drug delivery is a hot topic in the field. We have not studied the contribution of long-range signals out of brain tissue, as we focus on cell-to-cell communication between GB cells and neurons.

6) • The number of animals used for different treatment conditions is not always specified. In general, how are statistics performed? For example, in figure 4e,f, the authors should be running statistics on a PER ANIMAL basis, not on a per count basis (ie N=813). If sections from 6 flies were assessed, counts should be compared per animal

so n=6. This may change the significance of findings dramatically. This is true for analyses in virtually all figures.

We apologize for the lack of clarity. Now the minimum number of animals is specified for every case and all shown analyses throughout the manuscript are per animal. For instance, quantification of synapse number is per animal. We tried to clarify this in figure legends and Materials and Methods.

Regarding glial cell number (fig 4E) and glial membrane (fig 4f) that the reviewer mentions, the N=813 is a mistake: it should be N>12. We apologize for the error which has been corrected in the new figure. Also, in this figure we analyzed whole brains and not different sections of the same brain, so each data point corresponds to one animal.

Specific Concerns:

7) • How was 1-5 days determined to represent “early glioma” versus 5-10 days as “medium glioma”? How does this correspond to human disease progression?

The usual lifespan of a fly bearing glioblastoma is around 15 days at 29°C with a 70% of humidity. At this temperature flies and food within glass tubes (to monitor locomotor activity using DAMs) experience strong dehydration after 6 days. This causes premature death not induced by the GB. Therefore, we decided to divide the assays into three temporal segments that recapitulate the entire GB development process, and named them as early, medium and late. Given the huge difference in GB growth rate, it is virtually impossible to find similar stages between human disease progression and fly tumor development.

8) • Why was a power of 40 chosen as the cut off between rhythmic versus arrhythmic? Is this arbitrary based on your model.

In general, the threshold to consider an animal rhythmic is arbitrary, with no formal rules that we are aware of. For the circadian analysis we used the FaasX software, developed by prof Rouyer’s team. In such software, by default the cut off for rhythmicity contemplates a power of 40 and a width of 1.5 hours, which is the one we used in our analysis. In prof Rouyer’s papers, animals were considered rhythmic with a power above 20 with a width of 2 hours. We also performed the analysis in this way and found little differences. We have added a sentence in materials and methods to clarify this (line 540): “the FaasX 1.31 software arbitrarily assigned these thresholds by default. We also analyzed our data using more conventional criteria (as defined by Dr Rouyer’s lab; power >20, width >2 h), showing similar results.”

9) • Only a minority of animals are arrhythmic in early GB – is there any biological explanation for this variance?

As the tumor advances, the circadian locomotor activity of the animal is progressively compromised, nevertheless the response of each animal to the GB is quite variable. However, in the case of early GB there is still a significant number of flies with functional central pacemaker neurons and motoneurons, with no apparent phenotype in geotaxis response. This may count for the lower number of arrhythmic flies in early GB animals.

10) • Figure 1A versus 1E controls seems to have different period lengths. Is this true?

We thank the reviewer for noticing this. Actually, period lengths are not different but actograms shown in figures 1E-H did not match with depicted Light:Dark periods. This is the reason why the morning peak (the locomotor activity peak corresponding to dawn) apparently appeared one hour before light onset, indeed suggesting a difference in period lengths which is not real. We have substituted them by corrected actograms with adequate time plot and L:D regime (new figures F-I).

11) • Figures 1B and 1F are labeled as rhythmic but appear quite arrhythmic, especially compared to controls – significance values on these graphs would be helpful

To avoid confusion, we have changed them by two examples that are more representative. Following a suggestion of the other reviewer, we also changed the actogram representation

(from FaasX software) to a more readable (and classical) version (using the set of R-packages defined by Rethomics- <https://pubmed.ncbi.nlm.nih.gov/30650089/>) in all the figures, with locomotor activity depicted as shaded bars. We hope that new actograms are clearer in order to detect rhythmicity.

Regarding significance values, actograms shown in figures 1A-H and 3A-C are graphical representation of the locomotor activity of one particular animal, depicted as examples. In the text and figure legends we have tried to clarify it, i.e. that an actogram is just exemplifying the movement of one single fly. For the text we added the following sentences (lines 110 and 114) "Control flies were strongly rhythmic, as shown by an actogram depicting the locomotor activity of one single fly that was representative" and "Fig 1B-C illustrates two representative actograms of single early-GB flies as examples of rhythmic or arrhythmic animals." We described it in the figure legends as follows (lines 128 and 249): "Representative individual locomotor activity profile from one particular animal of each indicated genotypes showing 5 days under free running (D:D)..."

The mathematical analysis of rhythmicity for each actogram is performed with the periodogram, which indicates the period (when a particular locomotor activity repeats in time, shown as a peak) and the "power" (the area below the peak and a threshold). For instance, below you can find the actogram of the fly we have chosen for new fig 1G (i.e. former fig 1F, given that we added the requested missing *elav>rheb* control as new figure 1E -see below-) together with its periodogram. Despite the actogram (A) seems quite arrhythmic, the periodogram (B) shows a peak at approximately 24 hours that clearly overpasses the almost horizontal threshold line (arrow, with the measured area colored in grey; actual power of 121-rhythmic). In contrast, the fly number 3 has an actogram (C) that looks pretty rhythmic, however the periodogram (D) shows that this animal does not behave rhythmically (power of 32). This analysis was performed for each fly, thus rendering the values shown in new figure 1K (former fig 1l) and table1.

12) • MISSING CONTROL: Figure 1 should have an actogram and power values measuring the periodicity of non-GB-bearing flies with the *ELAV>rheb* rescue

We agree that this control is missing. We have performed the experiment for early and medium stages, and found no differences in circadian parameters when compared with control. The actogram is clearly rhythmic both in early and medium stages, with a power of 183.4 and 163.3, respectively. The period is similar to controls, 23.7 and 23.8 h for early and medium. We added actograms for early and medium-staged *elav>rheb* animals, being new figures 1E (early, i.e. 0-5 days-old) and 1J (medium, i.e. 5-10 days-old), as well as data in new figure 1K. In addition, we incorporated the analysis to table 1. We changed the nomenclature in the text and figure accordingly. The added text is as follows (line 155): "In contrast, both early and medium control ELAV>rheb flies behaved rhythmically, with a 23.7 and 23.8 endogenous period, respectively (see fig 1E, J for illustrative actograms of each phase). Also the average power value was high, up to 183.4 ± 19.8 for early flies and 163.3 ± 18.7 for medium ones (fig 1K and table 1)."

13) • Why did authors choose to quantify synapse number at 7 days old and 7 days post-tumor induction for Figure 2? Why not 10 when the phenotype would likely be more pronounced?

In all cases we quantified synapses at 7 days post-tumor induction (i.e. at 29°C) and not 7 days old, which actually was a mistake. We have corrected this in the text. We chose day 7 because the effect of GB is notorious and statistically significant in virtually all the parameters that we can quantify, together with a high number of alive individuals. In contrast, at day 10 the phenotype is more evident but survival might be seriously compromised, especially because of the great variability in tumor growth.

14) • It is unclear whether the decrease in PDF positive puncta is due to overall loss of neurons or the same number of neurons expressing less pdf-? Both? That other clock genes are by and large unaffected would suggest GB specifically affects PDF.

The number of pdf-expressing cells that contact with DN1 is only four (<https://pubmed.ncbi.nlm.nih.gov/10619432/>) and we did not detect any neuronal loss as shown in the figure below. We took pictures of PDF neurons in glioma and control flies, and the total number of cells that innervate DN1 is four in both cases. We can conclude that the effect on pdf puncta is caused exclusively by a reduction in *pdf* expression levels. Indeed, we see lower pdf protein levels of the soma in glioma animals. We agree with the reviewer, GB seems to have an impact solely on *pdf* gene whereas *tim/per* or *clk/cyc* remain unaffected (see figure 2).

15) • MISSING CONTROL: Why were the number of PDF punctae not quantified in control>rheb flies? If insulin signaling increases active zone number, it may also increase pdf punctae at base-line without glioma given, “the inter-dependence between the number of active zones and PDF protein levels: modifying either one affects the other” -Line 159-160. This could imply that insulin signaling is contributes, but is not

sufficient, to rescue pdf expression/rhythmicity; this is supported by PDF-rescued GB-bearing flies having similar period, power, and survival to GB-bearing flies

We apologize. Actually, we had the data but we thought that they would not be necessary. Insulin signaling does not increase significantly the active zone number and PDF punctae (although there is a tendency), thus suggesting that it is not sufficient to fully rescue pdf expression/rhythmicity. We included a confocal image of PDF neuronal contacts with DN1 stained with anti-*pdf* (green) from flies overexpressing *rheb* in PDF neurons as new fig 2J. Quantification of such contacts are now added in the new fig 2M (former 2L). We also modified the end of the paragraph, following another suggestion of the reviewer (see below), as follows (line 218): "PDF-rescued GB brains tended to increase the number of PDF puncta when compared to GB samples, showing no significant differences with control brains (fig 2K-M). Control flies that overexpressed *rheb* in PDF neurons had higher number of active zones/PDF contacts than control active zones/PDF contacts than control and PDF-rescued GB samples, however this difference was not significant. These results suggest that insulin pathway activation can partially rescue the loss of active zones in GB induced brains." If he/she does not mind, we would like to borrow the sentence from the reviewer. It is tentatively added at the end of the paragraph (line 226): "Insulin signaling contributes, but is not sufficient, to rescue *pdf* expression/rhythmicity."

16) • For Figures 4 and 5, are all glial cells in the membrane part of the tumor volume? Where are the tumor borders?

The membrane-targeted myristoylated RFP used to quantify membrane volume is expressed in all glial cells. Indeed, the same *repo-gal4* driver that induces tumor formation (by means of overexpression of *PI3K* and *EGFR* constitutively active forms) also is responsible of *myrRFP* expression. Thus, tumor borders are located where glial cells are in contact with the wild type neuronal population. We described details of such interphase between tumoral glial cells and neurons in a previous manuscript (see <https://pubmed.ncbi.nlm.nih.gov/31846454/>)

17) • It is unclear which data points the significance values apply to throughout the entirety of the manuscript.

We changed the way that significance between data values was represented along the manuscript in order to make them clearer.

18) • The N values throughout the manuscript are highly variable in between experiments and different treatment conditions.

The reviewer is right, the number of animals used in the different experiments changes greatly from one to another, mainly due to the type of performed tests. For instance, in locomotor activity or survival experiments, a large n (at least 32 or 80 animals, respectively, although some of them died during the trial) is required to avoid artifacts throughout the experiment. For others such as brain size and NMJ analysis experiments, a minimum of 8 animals are used per experiment, performed at least twice and with similar values. The exceptions are some samples of figure 2 with N values of 5-6, due to the difficulties to obtain the correct genotypes after several attempts (see Supplementary Data).

19) • While it may be intriguing to extend these findings to human glioma patients; implementation of a 14/14 LD cycle seems neither feasible nor accessible.

The referee raises an interesting point. Actually, our findings need to be validated in human GBM patients: in the discussion we proposed to study the evolution of circadian changes that accompanied GB progression. It is likely that some patients may have altered their own circadian rhythms in early GB stages (and not necessarily with a 14:14 period) that is probably disrupted in later stages. It is true that the implementation of a cycle other than 12:12 for long periods in human patients is not feasible, given that human daily routine is not fixed as it is in our experiments. We removed the idea of implementation of 14:14 regime for human patients and revised the abstract, removing the following sentence (line 23): "if such mechanisms are conserved, these results in flies may open new therapeutic approaches to extend lifespan of

glioblastoma patients through modulation of environmental cues." We also modified the main text, as follows (line 435): "the first step would be to determine whether or not the GBM mammalian model behaves similarly to Drosophila GB. This would require a detailed study of the circadian period of early diagnosed GB patients by measuring different parameters". We also added a brief discussion about sleep alterations in GB patients and their possible causes (line 390). In fact, glioma patients might benefit positively from a reinforcement of their circadian rhythms and sleep by following certain routines. We also incorporated this idea in the discussion (line 444): "in any case, it seems clear that the implementation of a cycle other than 12:12 for long periods in human patients is not feasible. Nevertheless, glioma patients might benefit positively from a reinforcement of their slightly altered circadian rhythms and sleep by following certain routines."

20) • Line 28 – Should read “glial cells”, not “glial cells”

We apologize but we do not understand this concern

21) • Line 68 – Should read “BMAL1”, not “BMLA1”

We apologize for the error. We corrected it in the text (line 72)

22) • Figure 1A-1H – axis labels are unclear; how many animals are represented in each graph? In general, figure quality is quite low and difficult to read

We are very sorry for the bad quality of figures. We improved them in the new version of the manuscript. Regarding panels 1A-H, each panel shows the locomotor activity of one single fly as a representative example. We clarified this in the text (lines 110 and 115) and figure legends (lines 128 and 249). The initial number of animals we used for each genotype is 32 animals, although some of them die during the experiment. It is stated in the figure legends.

23) • Figure 1I – Graph lacks a title and clear axis labels/units, unclear which data values are significant

The reviewer is totally right, the graph is puzzling. We added the following title "rhythmicity quantification". The Y-axis is the power value but the X-axis is confusing: we have renamed 0-5 and 5-10 as "early stages" and "medium stages", respectively; actually, it represents the power values of at least 22 flies with the genotypes depicted in A-E (early) and F-J). We also changed the way that significance between data values was represented to make it clearer. We sincerely hope that the graph is now comprehensible.

24) • Figure 1I does not have a figure legend

We apologize for the mistake. Fig 1I was mislabeled as 1g. We changed it to 1K in the panel and figure legend, given the two additional panels of the figure from the requested missing *elav>rheb* control.

25) • Figure 2F legend does not clearly state the age of flies when active zones were quantified

All brain and synapse number analyses are performed after 7 days of glioma induction. Revising the text we found that, by mistake, we used the term "7 days old" instead of "7 days post-tumor induction". We corrected it throughout the text.

26) • Figure 2L is confusing, how is that the insulin signaling rescue has no significant difference in PDF punctuae number with the control and no significant difference in PDF punctuae number with the glioma-bearing flies?

To compare more than two groups of data we needed to use an ANOVA, post-hoc Bonferroni test, which is more restrictive than a t-student test in order to identify significant differences. Actually, the ANOVA test only detected differences between control and glioma. However, when we compared glioma with pdf-rescued GB brains using a t-student test we did see significant

differences. To avoid confusion, we removed the sentence referring to t-student analysis and modified the text as follows (line 218): "PDF-rescued GB brains tended to increase the number of PDF puncta when compared to GB samples, showing no significant differences with control brains (fig 2K-M). Control flies that overexpressed *rheb* in PDF neurons had higher number of active zones/PDF contacts than control samples, however this difference was not significant. These results suggest that insulin pathway activation can partially rescue the loss of active zones in GB induced brains."

27) • Figure 3D – What day into glioma progression are these flies? The

The panel 3D represents the power values of medium-staged flies (i.e. from 5 to 10 days at 29°C, i.e. GB induction in the case of glioma), as stated in the figure legend. We also incorporated this information in the main text (line 234): "In particular, despite medium-staged (i.e., 5 to 10 days of tumor development) control and GB animals were both rhythmic..."

28) • Figure 3F lacks any p-values/significance and lacks a Y-axis label

GB flies started to show problems in geotaxis at day 7, with no movement at day 15. However, all wild type flies of the same age showed no phenotype, so we thought that differences were quite clear. Besides, this is a percentage, so p-values cannot be used. We also added the "geotaxis response (%)" labeling to the Y-axis.

29) • Figures 3G-3I are not legible

We deeply apologize. We improved the image quality in the new version of the manuscript.

30) • Figure 4 and 5 titles are inaccurate – 14/14 regime improves survival but there is no evidence that the tumor progression is impaired: only glial cell number is increased by LD 12/12, and tumor volume is the same between 12/12 and 14/14.

We have modified the title and text of figure legends 4 and 5. For figure 4 the new title is (line 319): "Synchronization of environmental light/dark phases with internal period prevents GB-induced neurodegeneration." For the figure 5 is (line 348): "Re-adjusting environmental cues to internal rhythms after GB induction reduces tumor proliferation." Indeed, in the main text this distinction was already made.

31) • In Figure 4, how are specifically glial membranes and nuclei labeled?

The membrane of tumor glial cells is marked with a membrane-targeted myristoylated RFP UAS-transgene expressed under the control of the repo-gal4 driver. We clarified this in Mat and Met (line 581): "Glial network was marked by a membrane-targeted myristoylated RFP UAS-transgene (*UAS-myr-RFP* reporter) specifically expressed under the control of repo-Gal4". The glial nuclei are marked by an anti-repo antibody.

32) • In Figure 4F, why is there any tumor volume for controls?

The Y-axis labeling should have been "glial volume (um³)", as in figure 5D. We corrected it in the new version.

33) • In Figures 4 and 5, it would be clearer if labels were 12/12 rather than LD as the 14/14 also experience light/dark cycles.

We thank the reviewer for the suggestion, which facilitates the reading and makes the text much clearer. We changed LD to 12/12 throughout all the text and figures.

34) • (p)50 is not explained in the figure legend

We added the following sentence in the figure legend to explain p50 (line 329): "p50 is defined as the time when half of GB flies are still alive respect to the time when the same proportion of"

control animals are dead, in percentage". We also made a mistake in the p50 calculation (as can be seen by looking at fig 4L) so we have corrected it in the new figure 4M.

35• Figure 5D Y-axis has strange numbering/labeling

We changed it to μm^3 , as in figure 4F.

Reviewer #2 (Remarks to the Author):

In this manuscript by Jarabo et al., the authors look at the interaction between glioblastoma-induced neurodegeneration and circadian rhythms in flies. The authors find that glioblastoma normally disrupts circadian rhythms in a majority of animals due to an increased prevalence of neurodegeneration. Rescuing neurodegeneration in all neurons rescues disrupted rhythms. While this may be due to changes to pdf-expressing neurons (the authors find glioma-induced changes in active zones and pdf expression), they surprisingly find that rescuing neurodegeneration in only pdf neurons has no effect on rhythmicity as a whole. However, pdf rescue does seem to rescue a glioma-dependent decrease in morning anticipation. Finally, and most interestingly, the authors find that 1) flies with glioma that are still rhythmic have lengthened periods and, 2) lengthening day length to a 14:14 LD cycle delays tumor progression and increases survival. Overall, the manuscript is novel and the statistics seem to be appropriate. The manuscript will be of interest to cancer biologists, chronobiologists, and developmental biologists. The manuscript will be improved when the authors address the following concerns:

Fig 1.

1) 1A-H: It would be helpful if the actograms were shaded or filled in so that the reader can more easily determine rhythmicity/arrhythmicity in each trace.

We thank the reviewer for the suggestion. We have changed such actogram representation (from FaasX software) to a more readable (and classical) version (from Rethomics package, see below) in all the figures, with locomotor activity depicted as shaded bars. We hope that new figures are clearer in order to detect rhythmicity.

2) 1I: Please define the X axis (days of tumor development?) and A. Please define the red line.

The X axis are the experimental genotypes shown in A-J, grouped in early and late stages. We renamed 0-5 and 5-10 as "early stages" and "medium stages", respectively, to clarify it. We also changed the way that significance between data values was represented to make it clearer. The red line marks the threshold to consider power values arrhythmic (i.e. 40). We explained this is the figure legend (line 136).

3) 1I: Please make it more clear what groups are significantly different from one another. It's difficult to tell with the authors' current layout.

We also changed the way that significance between data values was represented to make it clearer. We sincerely hope that the graph is comprehensible now.

4) Table 1: Was the period increase in glioma flies statistically significant?

We have analyzed all the experiments and we have not found statistically significant differences in any case. However, we have observed that the variance is statistically different in GB, suggesting higher differences that the behavior of each individual with GB is different (Supplementary data).

Fig 2.

5) 2H. The authors should show individual data points if possible

We thank the reviewer for the suggestion. Indeed, we tried to do so in the majority of figures. Survival (fig 3E and 4L) and locomotion (fig 3F) are shown as percentage, and actograms are representative examples of one single animal. Thus, individual data points of these graphs cannot be shown because they are already individual. The locomotor activity depicted in figure 3G-I uses the set of R-packages defined by Rethomics (<https://pubmed.ncbi.nlm.nih.gov/30650089/>). New information is now added in Materials and Methods (from now on, new text is underlined, line 547): "The solid lines and the shaded areas

show population means and their 95% bootstrap confidence interval, respectively." We tried to plot individual data points but the graph was too confusing. Regarding the qPCR (fig 2H), we showed now all data points in the new fig 2H.

6) 2H. The authors extracted RNA at ZT 6 -- was pdf constitutively low, or just at this time point?

pdf expression in glioma brains at ZT6 are low when compared to *pdf* expression of control ones, whereas the other circadian genes show similar expression levels. However, we are not aware of published differences in *pdf* expression throughout the day. In addition, ZT6 was the time point where all the dissection experiments were done in the manuscript.

Fig 3.

7) 3G-I: The figure is very pixelated and may need to have increased resolution for readability

We have included an improved version of the image in the revised manuscript.

8) 3G-I: Have you performed statistics to show that there is indeed a difference and that glioma + pdf insulin signaling is able to restore the morning anticipation peak?

The reviewer is totally right. We have thought very seriously about the statistical approach to solve this issue. However, the fact that the glioma+ pdf rheb rescue is progressive makes the analysis complex: in day 1 it is similar to glioma flies whereas in day 4 the glioma pdf-rheb rescued is similar to control ones. If we compare anticipation (defined arbitrarily as locomotion during 3.5 hours before the expected lights-on) day-by-day (which is not the usual way for circadian analysis) there are significant differences with control animals in day 1 and not in day 4. To do this analysis, we used a particular script from Rethomics package. We added this new information in the main text as follows (line 286): "in the first morning after GB (and PDF-rescued) onset, the MA of GB and PDF-rescued GB flies was indistinguishable and showed a significant difference with control animals; in contrast, the fourth day showed a peak of anticipation similar to control animals that is significantly different to glioma flies (fig 3H-I)." We also have added this information in new box plot graphs (figure 3H-I), together with a modified figure legend (line 261): "Detail of day 0.5-day 1.5 and day 3.5 and 4.5, showing no rescue and an almost full rescue of morning anticipation, respectively, together with the statistical analysis of the morning anticipation in day 1 and day 4. At day one, there is a significant difference between control and rescued glioma animals, comparable to glioma flies. At day 4, however, rescued glioma animals are more similar to control ones and significant different to glioma flies (Pairwise comparisons using Wilcoxon rank sum test) (*p-value < 0.05)."

9) Fig. 4

Why does a lengthened day decrease the total glial cell number in glioma flies to that of control flies, but not tumor volume?

GB cells number and tumor volume are two related features, but not interdependent. Actually, our own results (<https://pubmed.ncbi.nlm.nih.gov/33526430/>) showed that *miR-8* expression in GB could increase tumor volume without modifying glial cell number.

It is tempting to speculate about mechanisms underlying the effect of longer day/night periods on GB progression. A previously proposed chronogram for GB progression and its cellular mechanisms involved the expansion of GB membrane, which in turn would drive the exchange of signaling molecules between GB cells and neurons. Ligands such as WNT/Wingless or ImpL2 mediated GB progression and neurodegeneration. Our previous data suggested that impairment of cell-to-cell communication is sufficient to block GB progression. Our hypothesis is that a 14:14 L:D regime did not reduce the expansion of GB membrane, with GB structures such as Tumor Microtubules still present, whereas it could affect the signals that mediated cell to cell communication, and therefore glial cell number did not increase.

10) 4K: Are CLD vs. C14 and GLD vs. G14 significantly different based on post hoc tests?

Instead of increasing the number of active zones in the G14 condition a lengthened day could ostensibly be reducing the number of active zones in the C14 flies.

Indeed, G14:14 is significantly different from all other genotypes, while CLD, C14:14, and G14:14 are not significantly different from each other based on post hoc tests (Bonferroni).

11) The authors should comment on whether a 14:14 LD cycle increasing life span in wild-type flies is consistent with previous literature.

We thank the reviewer for the question. Actually, circadian desynchronization between the central clock and external cycle have a great impact in physiological and cognitive functions from flies to humans, with serious health consequences (reviewed in <https://pubmed.ncbi.nlm.nih.gov/30269400/>; see also <https://pubmed.ncbi.nlm.nih.gov/32814592/>). Besides, such misalignments accelerate aging and causes premature death in flies and primates (for instance, see <https://pubmed.ncbi.nlm.nih.gov/4624759/> and <https://pubmed.ncbi.nlm.nih.gov/33093578/>). In fact, mutant flies with a natural cycle of 29 h outnumbered wt flies (i.e. 24 h cycle) when incubated under a 14.5:14.5 L:D regime (<https://pubmed.ncbi.nlm.nih.gov/31736790/>). Altogether these data suggest that a correct synchronization improve general fitness and survival. However, all described experiments have been performed at 25°C and not at 29°C. At this high temperature for *Drosophila* (29°C) fly fitness is seriously challenged. Actually, both circadian and sleep patterns at 29°C are quite different from the ones at 25°C (besides our data, see <https://pubmed.ncbi.nlm.nih.gov/10677039/> and <https://pubmed.ncbi.nlm.nih.gov/31572216/>), which may account for the increased lifespan we see in our experiments at 29°C under a 14:14 L:D regime. However, further experiments are needed to fully address this issue. We added a comment in the revised discussion as follows (line 448): "A surprising result was that control animals in 14:14 L:D cycles lived considerably longer than their counterparts at 12:12 L:D. Data from flies and mammals suggested that a correct circadian synchronization improve general fitness and survival 44. However, in the case of flies all experiments were performed at 25°C, whereas in our case we used 29°C, a challenging temperature for *Drosophila* fitness. Indeed, both circadian and sleep patterns are quite different at such high temperature compared to the ones at 25°C 45,46, which may account for the increased lifespan we see in our experiments at 29°C under a 14:14 L:D regime."

Nevertheless, GB flies display a survival from 40% to 60% respect to wt flies (p50) while GB flies with a significant effective rescue show an increase in lifespan of at least 20% (see this manuscript -p50 of 57% and 77% of GB and rescued GB animals, respectively-, (<https://pubmed.ncbi.nlm.nih.gov/35216153/>) - p50 of 47% and 73% - and <https://pubmed.ncbi.nlm.nih.gov/33526430/> -64% and 88%-).

General:

12) Are glioma cells in flies rhythmic and, if so, do they show similar circadian periods to non-cancerous cells?

Given the influence of glia in circadian regulation (<https://pubmed.ncbi.nlm.nih.gov/32075519/>) we suspect that glioma cells are getting progressively arrhythmic, something that may contribute to the disruption of the fly circadian behavior. Actually, one of the proposed mechanisms by which glia controls central clock is through regulation of pdf protein levels (<https://pubmed.ncbi.nlm.nih.gov/21497088/>), although other data do not support it (for instance, <https://pubmed.ncbi.nlm.nih.gov/29487148/>). Regarding how to determine glial circadian rhythms, virtually all the work in *Drosophila* that shows the role of glial cells on central clock are mainly functional, with the exception of PER protein cycling (see <https://pubmed.ncbi.nlm.nih.gov/31840430/>). Interestingly, blocking glial PER function does not affect circadian behavior (<https://pubmed.ncbi.nlm.nih.gov/21497088/>). We did not perform any experiment trying to detect molecular rhythmicity in GB brains: the intrinsic variability of individual GB progression and the disease progression itself would difficult a precise quantification of PER protein levels. This would require the dissection and staining of several

brains without previous knowledge of the circadian behavior of each animal, thus making the experimental dataset hard to interpret.

Despite we do not see expression differences in the central core of circadian genes (such as *per/tim* and *clk/cyc*), we do have data indicating a disruption in the expression of another circadian gene besides *pdf*. Indeed, *cry* expression is increased 50 times in glioma cells. These results are included in a recently published manuscript (<https://pubmed.ncbi.nlm.nih.gov/35216153/>)

13) There are several grammatical/spelling errors throughout the manuscript. Please proofread it carefully prior to the next submission.

We have revised carefully the grammar and spelling in this new version.

MODIFIED FIGURES (for both reviewers)

Figure 1

We changed the visualization of actograms to a more readable version (Rethomics software). Panels 1B and 1H were changed for another example more representative. For panels 1F-H both the actogram and the L:D regime are now correctly adjusted. The *elav>rheb* control is added (new fig 1E and 1J) as well as their data in the panel 1K. The significance value representation was also modified and changed from 0-5d and 5-10d to early and medium stages.

Figure 2

The individual data points of panel 2H are now shown. The *pdf>rheb* control was added as an image (2L) and in the quantification (2M). Significance value bars between data were also modified.

Figure 3

We changed the actogram visualization to the Rethomics-derived ones, as well as the significance value bars between conditions. We added a statistical quantification of morning anticipation using again the Rethomics software.

Figure 4

We modified the significance value representation between data. We also modified changed L/D by 12/12

Figure 5

We change GC and GT labels to 12/12 and 14/14 to indicate the L:D regime. We labeled the Y axis as tumor volume(μm^3) in panel 5D. We modified the significance value representation between data.

Reviewers' comments:

Reviewer #1 (Remarks to the Author):

While the author have made numerous changes which strengthen their claims, including the addition of many missing controls and Table 1, there remain a few issues that should be addressed.

- 1) While the authors do highlight the important interactions between neurons and glioma cells, they do not address the real confound of a model in which all the glial cells are converted to glioma cells, leaving no healthy glia in the microenvironment. The authors should thoroughly address this confound in the discussion.
- 2) In relation to the model itself, the authors claim this model faithfully recapitulates what happens in human GBM because of its diffusivity. However, the authors should discuss that GBMs do not typically arise everywhere in the brain and there is a strong spatio-temporal pattern of their development.
- 3) The justification of PDF decrease as a decrease in PDF expression is not very convincing, especially related to the added images in point #14.
- 4) In general, many of the figures are still difficult to read. The formatting between figures is not consistent, many axes and fonts are impossible to read (especially Fig. 3G-1 and Fig. 2L is missing a label). This makes it hard to interpret the data. Fig. 4L seems unnecessary as those data are conveyed in Fig. 4M.
- 5) In point 28 of the rebuttal the authors claim you can't have significance depicted in Fig. 3F because they are percentages but then the authors have significance depicted in Fig. 3E. What is the difference?

Reviewer #2 (Remarks to the Author):

The authors have satisfactorily addressed my concerns. I endorse the manuscript for publication in Communications Biology.

Reviewer #1 (Remarks to the Author):

While the author have made numerous changes which strengthen their claims, including the addition of many missing controls and Table 1, there remain a few issues that should be addressed.

1) While the authors do highlight the important interactions between neurons and glioma cells, they do not address the real confound of a model in which all the glial cells are converted to glioma cells, leaving no healthy glia in the microenvironment. The authors should thoroughly address this confound in the discussion.

We would like to thank the reviewer for raising this concern; we agree that to specify the limitations of each experimental model used to obtain the results is of great relevance. In order to clarify this, we included in the last version of this manuscript the first lines of the discussion to clarify that “The current *Drosophila* model of GB cannot reproduce the initial steps of GBM, given that all glial cells are transformed, so the interaction between healthy and tumoral glial cells cannot be assessed...”. In addition, we included a list of features relevant for GBM aggressiveness and malignancy, reproduced in this *Drosophila* model.

It is true that the contribution of healthy glial cells, mainly astrocytes and microglia, to GBM progression is becoming more important in recent years. Indeed, it is described that “GBM cells can actually activate astrocytes, namely, the tumor associated astrocytes (TAAs), to promote GBM invasion in the healthy tissue” (<https://onlinelibrary.wiley.com/doi/10.1002/glia.23520>). Therefore, healthy glial cells promote GBM migration, invasion, proliferation and angiogenesis. (<https://www.frontiersin.org/articles/10.3389/fncel.2018.00235/full>). Unfortunately, whereas in mammalian brains 50% of cells are glia, in *Drosophila* the percentage is 10% (<https://cshperspectives.cshlp.org/content/7/11/a020552.long>), thus indicating that this *Drosophila* GB model might not be the most indicated to study GB-healthy glia interaction. We have included this information in the discussion (line 361 of the revised version).

2) In relation to the model itself, the authors claim this model faithfully recapitulates what happens in human GBM because of its diffusivity. However, the authors should discuss that GBMs do not typically arise everywhere in the brain and there is a strong spatio-temporal pattern of their development.

We agree with the reviewer, GBMs arise from a single cell in a clonal manner. This *Drosophila* model does not reproduce the origin and primary formation of GB, rather it aims to reproduce the infiltrative nature of GBMs, and the diffuse front that advance through the healthy tissue even after surgical removal of the core GBM cells.

We have clarified in the text that our intention is to study the neurological defects caused by GBM progression after surgery, and even after chemo and radiotherapy (line 370 of the revised version). To this end, this model reproduces a scenario in which GBM cells are distributed throughout the brain without generating a big core mass. This first approach contributes to the general understanding of GBM neurological damage, but it

will require further experiments with new models that induce GBM transformation in sub-populations of glial cells to include the contribution of healthy glia to tumor progression. We tried some strategies to induce the oncogenic transformation of unmyelinating glial cells (oligodendrocyte-like cells), and to generate clones of PI3K and EGFR pathways activation. However, the random localization of the clones prevented us from doing behavioural experiments as the reproducibility was very limited. In addition, the use of a specific glial sub-population raised novel concerns on the cell of origin for GBM.

Thus, we consider that, with limitations, the contributions of this model are relevant for the scope of this publication, and reach solid conclusions on the possible effect of GBM-induced neurodegeneration and the role of circadian rhythms.

3) The justification of PDF decrease as a decrease in PDF expression is not very convincing, especially related to the added images in point #14.

In addition to the qPCR showing lower *pdf* expression levels (a gene that is mainly expressed in pdf-expressing neurons- fig 2h) and the quantification of pdf puncta in PDFn-sLNv synaptic contacts (fig 2m), we have also quantified the intensity of fluorescence in the soma of pdf-expressing neurons both in control and GB brains. We observed that the mean of intensity is lower in GB brains than in control brains (59 ± 16 vs 28 ± 5 , $n > 8$, see also the graph below), with a difference that is significant (p value equals 0.0473). This suggests that pdf protein levels are reduced in GB PDF neurons. In addition, we do not see neuronal loss of sLNv-innervating neurons. All these data suggest that pdf decrease may be due to a reduction in pdf expression, although it is true that we have not designed a specific experiment to discard that it is caused by enhanced protein degradation or other alternative explanation.

4) In general, many of the figures are still difficult to read. The formatting between figures is not consistent, many axes and fonts are impossible to read (especially Fig.

3G-1 and Fig. 2L is missing a label). This makes it hard to interpret the data. Fig. 4L seems unnecessary as those data are conveyed in Fig. 4M.

We have re-formatted all the figures, increasing the font and homogenizing the formatting among them. We sincerely hope that they are clearer now. Despite the fig 4L might be redundant, a similar graph is depicted in another figure (fig 2, for instance). The fig 4M summarizes and simplifies the main finding of fig 4L, but we believe that the information provided by fig 4M might be of interest to the reader, so we would like to keep the figure as it is.

5) In point 28 of the rebuttal the authors claim you can't have significance depicted in Fig. 3F because they are percentages but then the authors have significance depicted in Fig. 3E. What is the difference?

We apologize for the lack of explanations. In figure 3E we applied the Mantel–Cox survival test to our data, that is indeed applied to clinical data in order to see differences in lifespan. We tried to apply it to fig 3F but data did not fit. However, we applied a Mann-Whitney U test for each point and renders a significant difference (*p-value< 0.05) within the shown interval. We added this information in the figure legend (line 258 of the revised version)

Reviewer #2 (Remarks to the Author):

The authors have satisfactorily addressed my concerns. I endorse the manuscript for publication in Communications Biology.

REVIEWERS' COMMENTS:

Reviewer #1 (Remarks to the Author):

The authors have appropriately responded to my concerns. The figures are significantly improved.